# Coupling chemical mutagenesis to next generation sequencing for the identification of drug resistance mutations in *Leishmania*

Arijit Bhattacharya[1], Philippe Leprohon[1], Sophia Bigot[1,2], Prasad Kottayil Padmanabhan[1], Angana Mukherjee[1], Gaétan Roy[1], Hélène Gingras[1], Anais Mestdagh[1], Barbara Papadopoulou[1,2] & Marc Ouellette[1,2]*

Current genome-wide screens allow system-wide study of drug resistance but detecting small nucleotide variants (SNVs) is challenging. Here, we use chemical mutagenesis, drug selection and next generation sequencing to characterize miltefosine and paromomycin resistant clones of the parasite *Leishmania*. We highlight several genes involved in drug resistance by sequencing the genomes of 41 resistant clones and by concentrating on recurrent SNVs. We associate genes linked to lipid metabolism or to ribosome/translation functions with miltefosine or paromomycin resistance, respectively. We prove by allelic replacement and CRISPR-Cas9 gene-editing that the essential protein kinase CDPK1 is crucial for paromomycin resistance. We have linked CDPK1 in translation by functional interactome analysis, and provide evidence that CDPK1 contributes to antimonial resistance in the parasite. This screen is powerful in exploring networks of drug resistance in an organism with diploid to mosaic aneuploid genome, hence widening the scope of its applicability.

[1] Division of Infectious Disease and Immunity, CHU de Québec Research Center, Quebec City, Quebec, Canada. [2] Department of Microbiology, Infectious Disease and Immunology, University Laval, Quebec City, Quebec, Canada. *email: Marc.Ouellette@crchul.ulaval.ca

The protozoan parasite *Leishmania* is endemic in several parts of the World and remains a serious public health issue with an estimated 700,000–1 million new cases. Treatment of the disease relies primarily on chemotherapy with four drugs currently in use, namely pentavalent antimonials (Sb(V)), miltefosine (MIL), amphotericin B (AMB), and paromomycin (PMM). However, the efficacy of each chemotherapeutic intervention is getting restricted by toxicity, cost, access, and by growing drug resistance[1].

Whole-genome analysis of drug resistant *Leishmania*, first with comparative genomic hybridization, revealed that copy number variation (CNV) and change in ploidy were associated with resistance[2,3]. The use of next generation sequencing (NGS) has confirmed the importance of CNVs in resistance in *Leishmania* but has also indicated the prevalence of single-nucleotide variants (SNVs)[4–6]. More recently, the power of whole genome gain- and loss- of function screens for drug resistance studies have emerged. One such gain of function technique combining cosmid- or plasmid-based functional cloning and NGS accentuated the discovery of drug targets and resistance mechanisms in *Leishmania*[7]. The development of RNA interference (RNAi) target sequencing (RIT-Seq) in *Trypanosoma brucei* led to a powerful loss of function screen for genes linked to drug action or resistance;[8] however, *Leishmania* lacks RNAi machinery[9]. CRISPR-Cas system has been implemented effectively in *Leishmania*[10,11] which will eventually allow for genome-wide loss of function screens as demonstrated in mammalian cells[12,13]. These techniques, whether based on gain- or loss of function usually will not pinpoint SNVs, which include single nucleotide polymorphisms (SNPs) and small insertions or deletions (indels), an important contributor of drug resistance in most organisms. To address this limitation we coupled chemical mutagenesis to NGS. Even if *Leishmania* parasites possess mosaic aneuploid genomes, chemical mutagenesis has indeed been exploited in the past for generating drug resistant mutants[14,15].

Recently, chemical mutagens such as *N*-ethyl-*N*-nitrosourea (ENU) or ethyl methanesulfonate (EMS) were used in genome-wide screens where NGS helped in pinpointing drug resistance mutations in mammalian cancer cells[16]. Similar screens combining chemical mutagenesis and NGS, an approach called Mut-Seq[17], to probe for essential genes of phenotypes have been described in organisms spanning from bacteriophages to embryonic stem cells[17–21]. In this study, we have optimized the Mut-Seq procedure for *Leishmania* while selecting for MIL and PMM resistance. We have highlighted several candidate resistance genes by sequencing the genomes of 41 clones. By concentrating on recurrent mutations we have proven experimentally the role of many genes, harboring the mutations, in resistance to either MIL or PMM. We have also performed extensive mechanistic studies on the role of a protein kinase involved in PMM resistance.

## Results

### Generation of *Leishmania* mutants by chemical mutagenesis.
We used four different mutagens ENU, EMS, MMS (methyl methanesulfonate), and HMPA (hexamethylphosphoramide) against a freshly picked *L. infantum* clone while optimizing the mutagen concentrations, exposure (6–8 h) and recovery (24–36 h) times, and drug selection dose for both MIL and PMM (see also the Methods section). The results are summarized in Supplementary Table 1 and a total of 16 and 25 colonies growing respectively on MIL and PMM containing plates were individually grown and their $EC_{50}$ measured. All the mutagenized clones were between 2.5 to 8.5-times more resistant to either MIL or PMM in comparison to the parental wild-type cell (Fig. 1a).

### Next generation sequencing of the mutants.
The genomes of the resistant clones along with the wild-type parent were sequenced by NGS. Sequence reads were aligned to the 33-Mb *L. infantum* JPCM5 reference (version 8.0). The genome fold coverage for each mutant sequenced was between 35- and 100-fold (Supplementary Fig. 1a, b). *Leishmania* resists drugs through either SNVs or CNVs[22]. CNV analysis did not reveal specific locus amplification or deletion across mutants, with one exception, but did highlight changes in ploidy of chromosomes in independent MIL resistant mutants (chromosome 9, 12, 13, 22, 23, 26, and 31) and PMM-resistant mutants (chromosome 2, 12, 22, 23, 26, 31, and 32) (Supplementary Fig. 1c, d). The exception noted above was the deletion of a small locus on chromosome 6 that was observed for all 25 PMM mutants (Supplementary Fig. 2a). This deletion was confirmed by PCR in the three mutants tested (Supplementary Fig. 2b). This locus contains the ABCG1 and ABCG2 genes[23] that are associated with multiple activities[24]. The co-transfection of ABCG1-2 in mutant PMM25 modestly resensitized cells to PMM (Supplementary Fig. 2c). A similar deletion was also detected in MIL5 (Supplementary Fig. 2a), which showed $1.55 \pm 0.03$ fold resistance to PMM compared to the wild-type source clone ($p = 0.0004$, unpaired two-tailed t-test). SNPs and indels were also detected in several genes whose location and occurrence are represented on a 36 chromosomes map (Fig. 1b). The distribution of mutations between coding and intergenic regions was similar (Supplementary Fig. 3a) but we focused mostly on SNVs in coding regions.

The 16 MIL resistant mutants revealed between 384 to 670 and 10–21 non-synonymous SNVs/genome induced respectively with EMS and HMPA (Supplementary Fig. 3b). Synonymous SNVs were also observed but not further studied (Supplementary Fig. 3b). Most SNVs in MIL resistant mutants were heterozygous and only 26 were homozygous (Supplementary Table 2). Analysis of the mutations revealed that EMS induced mostly transitions while HMPA induced mostly indels (Supplementary Fig. 3c). The comparison of the 25 PMM-resistant mutants revealed between 440–595, 82–151, and 14–44, EMS-, ENU-, and MMS-induced non-synonymous SNVs/genome (Supplementary Fig. 3b). Similarly to MIL resistant mutants, SNVs were also found in non-coding sequences (Supplementary Fig. 3a), and synonymous SNPs were also present (Supplementary Fig. 3b). The majority of the mutations were heterozygous with only four PMM-specific homozygous mutations (Supplementary Table 3). As for the MIL screen, EMS induced mostly G/C to A/T transitions, ENU and MMS-induced spectrum of transitions and transversions, and indels were more frequent in the MMS derived populations (Supplementary Fig. 3c).

There were 1171 and 1597 genes with SNVs in at least two independent mutants for MIL (Supplementary Data 1) and PMM (Supplementary Data 2), respectively. We first prioritized homozygous SNVs. Second we selected genes mutated in at least five independent mutants as this recurrence could be linked with the resistance phenotype. Third we selected the genes that are functionally relevant in view of existing knowledge of drug action.

### Testing of mutations contributing to miltefosine resistance.
Out of the 16 MIL resistant mutants, the MIL transporter MT (LinJ.13.1590)[25] was mutated in 11 mutants (Supplementary Data 1). One of the SNVs was homozygous (W269* in MIL7) (Supplementary Table 2), the others were heterozygous and distributed along the entire coding sequence. Some were non-sense mutations (Table 1 and Supplementary Data 1). Mutants with either a non-sense mutation or with multiple mutations in MT generally displayed higher level of resistance. Transfection of the wild-type copy of the *MT* gene in mutants MIL7, MIL8, and

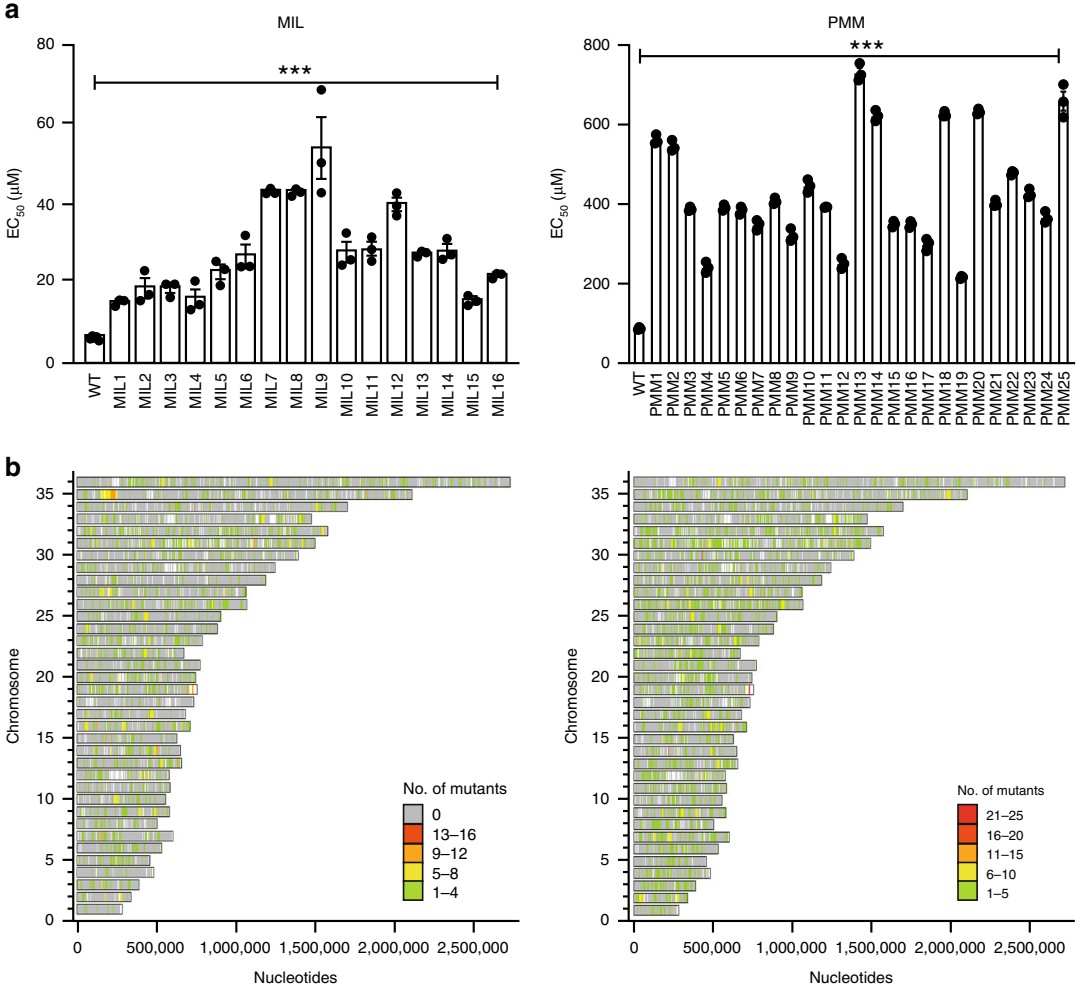

**Fig. 1 Drug susceptibility and mutations in *Leishmania* selected for resistance. a** Susceptibility to miltefosine (MIL; left panel) and paromomycin (PMM; right panel) were performed on individual clones. The wild-type *L. infantum* (WT) is shown for both drugs. The MIL resistant mutants were selected after mutagenesis with either EMS or HMPA (Supplementary Table 1) while the PMM-resistant mutants were selected after mutagenesis with EMS, ENU, or MMS (Supplementary Table 1). Data are mean ± SEM. For the MIL susceptibility assay, $n = 5$ biologically independent replicates for the wild-type and $n = 3$ biologically independent replicates for the mutants. For the PMM susceptibility assay, $n = 3$ biologically independent replicates. Statistical analyses were performed using unpaired two-tailed *t*-tests. ***$P < 0.001$. **b** Genome-wide distribution of SNVs in mutants selected against MIL (left panel) and PMM (right panel). Bars represent the genes on each chromosome. Colored bars represent genes mutated in defined numbers of mutant clones; gray bars represent non-mutated genes. Source data are provided as a Source Data file.

MIL9 resulted in a 3- to 5-fold resensitization to MIL (Table 1). Homozygous mutations in the gene *LinJ.30.1310* coding for a pyridoxal kinase were observed in four mutants (Supplementary Table 2). This gene has been proven previously to be associated with MIL susceptibility[5]. Another gene with homozygous mutations in MIL10 was *LinJ.36.6220* coding for a putative glycerophosphoryl diester phosphodiesterase (Supplementary Table 2). Transfection of the wild-type copy of *LinJ.36.6620* in MIL10 led to a low but significant re-sensitization (Table 1). Similarly, a gene coding for a fatty acid elongase, *LinJ.14.0790*, with homozygous mutations in 4 different mutants (Supplementary Table 2) was transfected in MIL6 and was found to resensitize the mutant to MIL (Table 1). The gene *LinJ.13.0300* coding for a long-chain fatty acyl CoA ligase with heterozygous mutations in 5 different mutants (Supplementary Data 1) was transfected in MIL15 and resensitized the mutant to MIL (Table 1). Episomal expression of the mutated versions of the long-chain fatty acid CoA ligase (*LinJ13.0300*) or of *MT* had no impact on the susceptibility of the mutants to MIL, while transfection of the mutated version of *LinJ.36.6220* bestowed further resistance to MIL (Table 1). Note

that other mutated genes might also be involved in MIL resistance but these were not tested. For example, the gene for the β-subunit (*ROS3, LinJ.32.0540*) of the MT[25] was mutated in MIL4 and MIL8 (Supplementary Data 1). It is salient to point out that we detected, upon sequencing of cells selected in a step-by-step fashion for MIL resistance, mutations in some of the genes highlighted by the Mut-seq screen (Table 1).

While our focus was on coding sequences, we looked into the mutations in untranslated regions that could theoretically change mRNA levels, as indeed *Leishmania* is known to control its gene expression by post-transcriptional or translational mechanisms implicating untranslated regions[26] although in the context of drug resistance this is usually achieved through changes in CNVs[2,27,28]. We selected five genes in MIL mutants with recurring SNVs in intergenic regions in more than one mutant (Supplementary Table 4). The RNA expression of three of those genes were changed (at most ~2-fold) and correlated with the SNVs. One gene was *LinJ.14.0340* that was upregulated by 1.7-fold in MIL15 (Supplementary Table 4). Transfecting the *LinJ.14.0340* wild-type gene in *L. infantum* wild-type cells did

**Table 1 Functional testing of selected genes in relevant miltefosine and paromomycin resistant mutants.**

| Drug | GeneID[a] | Mutants[a] | Mutation[b] | Type[c] | Fold resensitization WT gene[d,e] | Fold resensitization mutated gene[d,e] |
|------|-----------|-----------|-------------|---------|-----------------------------------|----------------------------------------|
| MIL | LinJ.13.0300 | MIL15 | E620Q | +/M | 1.41 ± 0.11* (n = 3) | 0.98 ± 0.08 (n = 6) |
| | LinJ.13.1590[f,g] | MIL7 | W269* | M/M | 3.36 ± 0.53*** (n = 6) | 1.02 ± 0.09 (n = 3) |
| | | MIL8 | D918N | +/M | 4.98 ± 0.42*** (n = 3) | ND |
| | | | S855L | +/M | | ND |
| | | | R544C | +/M | | ND |
| | | MIL9 | Q254* | +/M | 3.70 ± 1.29** (n = 3) | ND |
| | LinJ.14.0790[g] | MIL6 | FS155 | M/M | 1.81 ± 0.20* (n = 5) | ND |
| | LinJ.36.6220 | MIL10 | FS46 | M/M | 1.41 ± 0.28* (n = 6) | 0.65 ± 0.03** (n = 3) |
| PMM | LinJ.15.1490 | PMM7 | T44I | +/M | 1.64 ± 0.44** (n = 4) | ND |
| | LinJ.22.0640[f] | PMM7 | P250L | +/M | 2.15 ± 0.17*** (n = 3) | ND |
| | LinJ.27.1660 | PMM18 | L341F | M/M | 1.92 ± 0.34 *** (n = 4) | 0.96 ± 0.12 (n = 4) |
| | LinJ.28.2090 | PMM3 | S251F | +/M | 1.68 ± 0.1*** (n = 3) | ND |
| | LinJ.28.2390 | PMM6 | A76V | +/M | 1.61 ± 0.02*** (n = 3) | ND |
| | | | T415I | +/M | | ND |
| | | PMM8 | H123Y | +/M | 1.95 ± 0.06*** (n = 3) | 1.07 ± 0.15 (n = 3) |
| | LinJ.32.1830 | PMM20 | C70G | +/M | 1.64 ± 0.23* (n = 3) | ND |
| | LinJ.33.1810[f] | PMM25 | V366E | M/M | 4.61 ± 0.74*** (n = 6) | 1.01 ± 0.11 (n = 3) |
| | | PMM5 | W290* | +/M | 1.75 ± 0.15*** (n = 3) | ND |
| | | PMM16 | Q263* | +/M | 2.01 ± 0.55*** (n = 3) | ND |
| | | | G39D | +/M | | ND |

[a]A wild-type copy of the genes was expressed episomally in the mutants mentioned and drug responsiveness was assayed. LinJ.13.0300, long-chain-fatty-acid-CoA ligase putative; LinJ.13.1590, miltefosine transporter; LinJ.14.0790, fatty acid elongase; LinJ.15.1490, glutaminyl-tRNA synthetase putative; LinJ.22.0640, 50S ribosome-binding GTPase putative; LinJ.27.1660, Tubulin cofactor C domain-containing protein 1; LinJ.28.2090, elongation factor 2G mitochondrial putative; LinJ.28.2390, cyclin dependent kinase-binding protein putative; LinJ.32.1830, pumilio/PUF RNA binding protein 7 putative; LinJ.33.1810, protein kinase (CDPK1); LinJ.36.6220, glycerophosphoryl diester phosphodiesterase putative
[b]Mutations found in the mutants for the tested genes. The numbers indicate the position of the amino acid substitutions in the proteins. FS, frameshift = occurring at the amino acid position indicated.
[c]Mutation status of the mutants for the tested genes. M/M homozygous mutation; +/M, heterozygous mutation (+ is wild-type and M is mutation)
[d]Level of resensitization of the mutants upon episomal expression of the wild-type or mutated versions of the genes. Expressed as the ratio of EC$_{50}$ between the mutants with empty vector and mutant with the target gene. ND, not done. Statistical analyses were performed using unpaired two-tailed $t$-tests. ***$P < 0.001$; **$P < 0.01$, *$P < 0.05$,
[e]Source data are provided as a Source Data file
[f]Mutations in these genes were also observed in specific mutants of *L. major* that were selected in vitro directly for resistance to either miltefosine or paromomycin
[g]Mutation in these genes were also observed in specific mutants of *L. infantum* selected in vitro directly for resistance to miltefosine

increase the mRNA but had no effect on MIL susceptibility (Supplementary Table 4). Another gene was *LinJ.34.0690* in MIL6 that was downregulated by 2.3 fold. Transfecting the wild-type *LinJ.34.0690* gene in MIL6 slightly but significantly sensitized the mutant to MIL (by 1.6 fold) (Supplementary Table 4). We performed similar work for four genes in PMM mutants, for two we observed a modest change in mRNA levels associated with the SNVs, we studied *LinJ.14.0080* and *LinJ.35.0790* further but found no role in drug susceptibility (Supplementary Table 4).

**Testing of mutations contributing to paromomycin resistance.** We first concentrated on the role of the homozygous mutation V366E found in LinJ.33.1810 (CDPK1) in PMM25 (Supplementary Table 3). Indeed, in addition to PMM25, heterozygous mutations in CDPK1 were observed in 14 independent mutants including non-sense mutations in mutants PMM5 and PMM16 (Table 1 and Supplementary Data 2). A mutation in CDPK1 had also been observed in a *L. major* mutant selected for resistance to PMM in a stepwise fashion (Table 1). CDPK1 has a conserved N-terminal protein kinase domain (determined to be AMPK/CaMK by NCBI-CD-search) linked to an EF-hand-calcium sensor motif (Supplementary Fig. 4a). Similar kinases are found in plants and Apicomplexa parasites and are involved in a plethora of signaling activities in response to stresses[29,30]. A phylogenetic analysis suggests that the *Leishmania* CDPK1 is closer to the CDPK6 of Apicomplexa (Supplementary Fig. 4b). Episomal expression of the *CDPK1* gene (*LinJ.33.1810*) in the homozygous mutant PMM25 led to 4.6-fold re-sensitization to PMM and to 1.7- to 2-fold re-sensitization in the heterozygous mutants PMM5 and PMM16 (Table 1). Besides CDPK1, a gene coding for the tubulin binding cofactor C-1 (*LinJ.27.1660*) was also detected with a homozygous mutation (L341F) in PMM18 (Supplementary Table 3) which was resensitized up to 1.9-fold by expressing the wild-type gene (Table 1). In parallel we tested genes that were mutated in multiple independent mutants and whose function suggested a possible link with PMM action. The genes coding for a ribosomal GTPase (*LinJ.22.0640*), a glutamyl tRNA

synthetase (*LinJ.15.1490*), a cyclin dependent kinase binding protein (*LinJ.28.2390*), a mitochondrial elongation factor 2G (*LinJ.28.2090*) and one pumillio-RNA binding protein (*LinJ.32.1830*) could bestow at least 1.6-fold resensitization to PMM when wild-type genes were introduced into mutants in which the gene was mutated (Table 1). None of the mutants were sensitized to PMM following the episomal expression of the mutated version of the three genes tested (Table 1). Similar to the MIL mutants many additional mutated genes listed in Supplementary Data 2 may also be associated with the activity of PMM.

**The protein kinase CDPK1 and drug resistance.** By analogy to related orthologues[31], a lysine (K61) of CDPK1 should be critical for kinase activity and we thus generated a K61G mutant version. HA-epitope tag was added to wild-type and K61G versions of CDPK1 and these were expressed in *L. infantum*. We immunoprecipitated the CDPK1 kinase using an anti-HA antibody and the immunoprecipitate was incubated with γ-P$^{32}$-ATP. Multiple bands of varied intensity were detected from the immunoprecipitate of wild-type CDPK1 (Fig. 2a, lanes 2, 4), and cold ATP competed this phosphorylation (Fig. 2a, lane 3). In contrast, the kinase activity of the CDPK1$^{K61G}$ version was greatly impaired (Fig. 2a, lane 5). Transfection of wild-type *CDPK1* in PMM25 reversed resistance to PMM but not the transfection of the K61G version (Fig. 2b). We further studied the *Leishmania* CDPK1 gene by homologous recombination mediated gene deletion (Fig. 3a). We constructed *HYG*- and *PURO-CDPK1* inactivation cassettes (Fig. 3a). Southern hybridization indicated the proper integration of the *HYG* cassette in the *CDPK1$^{+/H}$* (+refers to the WT allele, H to the *HYG*-inactivated allele) line (Fig. 3b, lane 2), and similarly the *PURO* cassette was successfully integrated by homologous recombination in wild-type cells (Fig. 3b, lane 3) or in *CDPK1$^{+/H}$* cells, but in the latter a wild-type allele remained intact (Fig. 3b, lane 4), a phenomenon often observed for genes reputed to be essential in *Leishmania*. We have cloned the *CDPK1* ORF in the vector pSP72αZEOα which was transfected in the *CDPK1$^{+/H}$*-single knockout cells. Upon electroporation of the

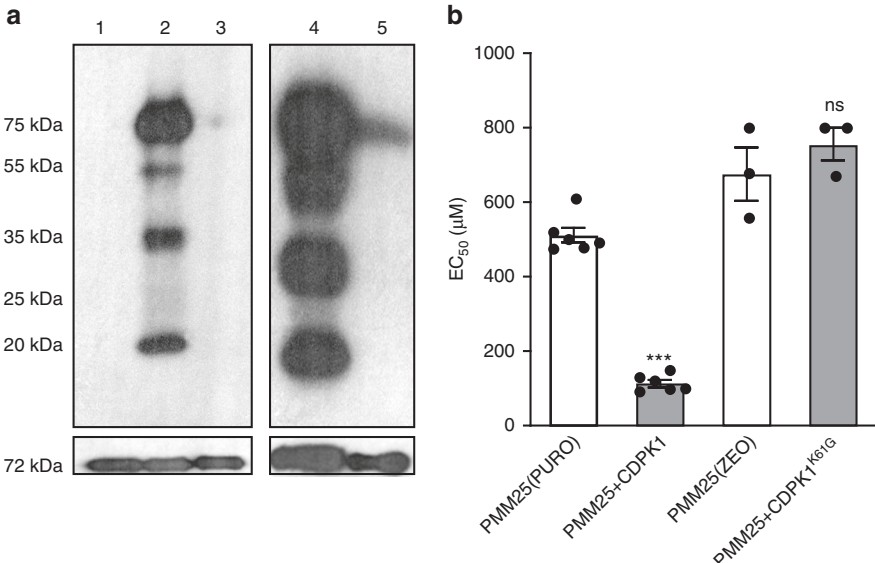

**Fig. 2 Kinase activity of the *L. infantum* CDPK1. a** Phosphorylation of proteins immunoprecipitated with CDPK1-HA as determined by autoradiography after in vitro kinase assay followed by electrophoresis in 12% SDS–PAGE. Assays were performed without γ-P$^{32}$-ATP (lane 1), with 2.5 μCi γ-P$^{32}$-ATP (2) and with γ-P$^{32}$-ATP in presence of excess of non-radioactive ATP (3). In an independent set of experiments, assays were performed with immunoprecipitated CDPK1-HA (4) and CDPK1$^{K61G}$-HA (5) both using 2.5 μCi of γ-P$^{32}$-ATP. Lower panels depict the amount of immunoprecipitated CDPK1 involved in each reaction as observed by Western blotting of the immunoprecipitated complex using mouse anti-HA IgG. **b** Impact of the CDPK1 K61G mutation on PMM resistance. EC$_{50}$ values were determined by dose responsive curves against PMM. The wild-type and K61G CDPK1 versions were cloned in PURO or ZEO vectors respectively, thus explaining two PMM25 controls transfected with the empty PURO and ZEO vectors. Data are mean ± SEM for $n = 6$ (PMM25(PURO); PMM25 + CDKP1) or $n = 3$ (PMM25(ZEO); PMM25 + CDPK1$^{K61G}$) biologically independent replicates. Statistical analyses were performed using unpaired two-tailed *t*-tests. \*\*\*$P < 0.001$. ns, not significant. Source data are provided as a Source Data file.

PURO inactivation cassette we could indeed obtain a chromosomal null mutant of CDPK1 when an episomal CDPK1 rescue plasmid was present (Fig. 3b, lane 5). The integration of the cassettes and the absence of a chromosomal copy of CDPK1 in the rescued CDPK1$^{H/P}$ (P refers to the PURO-inactivated allele) were also confirmed by PCR (Supplementary Fig. 5a). The episomal rescue vector carrying CDPK1 was maintained after more than 50 passages in the CDPK1$^{H/P}$ chromosomal null cells (Supplementary Fig. 5b, c) while it was lost from the CDPK1$^{+/H}$ line suggesting that CDPK1 is indeed essential (Supplementary Fig. 5b, d).

Growth of the CDPK1$^{+/H}$ promastigote cells was similar to wild-type cells (Supplementary Fig. 6a) but these were three times less susceptible to PMM (Fig. 3c). Episomal add-back of CDPK1 reverted the phenotype (Fig. 3c). While most of our work was done with the more tractable promastigote stage, we also studied the survival of wild-type, CDPK1$^{+/H}$ and CDPK1$^{+/H}$ add-back cells in PMA-differentiated THP-1 macrophages treated or not with PMM. The infectivity of the three cell lines were comparable but the CDPK1$^{+/H}$ cells were more resistant to PMM in comparison to control cells or to the add-back CDPK1$^{+/H}$ cells (Fig. 3d). To probe whether the CDPK1$^{+/H}$ line responded differently to stress, we monitored the expression of a number of stress proteins upon heat stress. The expression of HSP70, HSP60, and BIP were similar between wild-type and CDPK1$^{+/H}$ promastigotes and increased expression of HSP60 and 70 were similar in both cell types upon heat induction (Supplementary Fig. 6b).

We used a CRISPR-Cas9 strategy for introducing the V366E mutation in the CDPK1 gene in wild-type *L. infantum* expressing Cas9 or alternatively for editing the CDPK1$^{V366E}$ to a wild-type version in PMM25 expressing Cas9 (Supplementary Fig. 7a). We co-transfected a ZEO cassette interrupting the PTR1 gene

(LinJ.23.0310) along with the guide RNAs targeting CDPK1 (Supplementary Fig. 7a). Cells growing in the presence of zeocin were presumed to have an edited CDPK1 gene, and this was confirmed by Sanger sequencing (Supplementary Fig. 7b). Wild-type cells with a CDPK1$^{V366E}$ version were 3-fold more resistant to PMM compared to the control cells (Fig. 3c) while the PMM25 mutant engineered to code for a wild-type version of CDPK1 was more sensitive to PMM (Fig. 3c). The same strategy of DNA editing using co-transfection with a ZEO marker targeting PTR1 was also used for the fatty-acid elongase LinJ.14.0790 involved in MIL resistance (Table 1). A PCR fragment containing the frameshift in LinJ.14.0790 of mutant MIL6 (Table 1) was successfully introduced in one allele of the edited cells. This heterozygous mutant was 1.5-fold more resistant to MIL (Supplementary Fig. 7d). This strategy applied to MT allele from MIL7 also confirmed the role of the W269\* mutation in MIL resistance (Supplementary Fig. 7d).

Intriguingly, through NGS of the genome of one *L. infantum* antimony resistant mutant we had noticed a E629K mutation in CDPK1 (ref. [32]). This mutation appears to alter CDPK1 function since the episomal transfection of CDPK1$^{E629K}$ could only partly revert the PMM resistance phenotype in the CDPK1$^{+/H}$ line (Fig. 4a). Significantly, the CDPK1$^{+/H}$ line was cross-resistant to SbIII, and this could be reversed by the episomal add-back of CDPK1 (Fig. 4b). This cross-resistance was specific to SbIII because the susceptibility to MIL of the CDPK1$^{+/H}$ line was similar to wild-type cells (Fig. 4c). In ongoing work we have mutagenized *Leishmania* cells with EMS and ENU that were selected for SbIII resistance. The CDPK1 protein was mutated in 22 out of 32 clones. The mutations are spread along the length of the protein and none of those mutations overlapped with the one identified during the PMM selection except a single non-sense mutation at W290 (Fig. 4d).

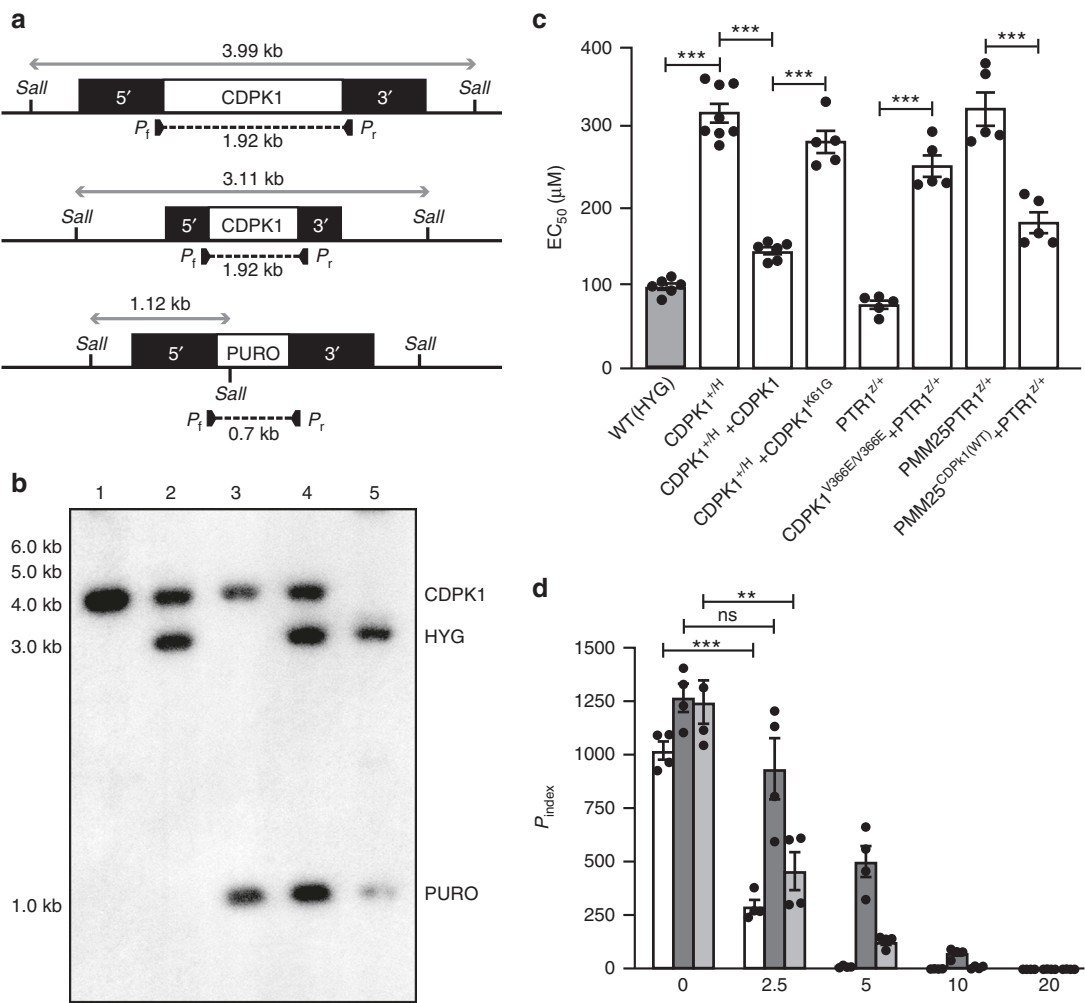

**Fig. 3 The essential *L. infantum* CDPK1 and its role in paromomycin resistance. a** Schematic representation of the CDPK1 locus in *L. infantum* before and after integration of the inactivation cassettes hygromycin phosphotransferase B (5′-HYG-3′) and puromycin phosphotransferase (5′-PURO-3′) and expected sizes after digestion with SalI. PCR primers ($P_f$ and $P_r$) and expected size are shown below the map. **b** Southern blot of DNAs digested with SalI and hybridized with a probe covering 300 bp of the 5′-UTR of *CDPK1*. *L. infantum* wild-type strain (1); $CDPK1^{+/H}$ (2); $CDPK1^{+/P}$ (3); $CDPK1^{H/P}$ (4) and $CDPK1^{H/P}$ + CDPK1 at passage 3 (5). **c** $EC_{50}$ values of *L. infantum* cells with various versions of CDPK1 were determined by dose responsive curves against PMM. Data are mean ± SEM for $n = 8$ ($CDPK1^{+/H}$), $n = 6$ (WT(HYG); $CDPK1^{+/H}$ + CDPK1) or $n = 5$ (all other samples) biologically independent replicates. Statistical analyses were performed using unpaired two-tailed *t*-tests. ***$P < 0.001$. **d** Enumeration of activated macrophage (THP1) infectivity and dose responsiveness of wild-type (white) and $CDPK1^{H/+}$ recombinant parasites supplemented (light gray) or not (dark gray) with an episomal wild-type copy of *CDPK1*, as determined by calculating $P_{index}$ after 96 h of PMM treatment. Superscript *H* and *Z* refer to the *HYG* and *ZEO* selectable markers, respectively. Superscript + refers to the wild-type allele. Data represents mean ± SEM for $n = 4$ independent biological replicates. Statistical analyses were performed using unpaired two-tailed *t*-tests. ***$P < 0.001$; **$P < 0.01$; ns, not significant. Source data are provided as a Source Data file.

**The CDPK1 interactome and prediction of putative targets**. Protein extracts of *Leishmania* expressing CDPK1-HA, CDPK1^K61G-HA, or CDPK1^V366E-HA were immunoprecipitated with an anti-HA antibody and these precipitates were subjected to LC-MS/MS analysis. We also carried out independent immuno-precipitation of HA-PTR1 and HA-DHFR-TS that were used for filtering proteins to concentrate on those interacting more specifically with CDPK1 (Average probability, AvP > 0.80). Three biological replicates were done for each sample (Supplementary Data 3). We focused on proteins either specific to the wild-type version or interacting with the three CDPK1 versions. Spectral counts from the identified prey-proteins were analyzed by SAINT[33] and the resulting interactome was visualized using cytoscape (Fig. 5a and Supplementary Fig. 8). Construction of a merged network with candidate interactomes of CDPK1, CDPK1^K61G-HA and CDPK1^V366E-HA indicated 18 unique prey proteins for CDPK1 (Supplementary Table 5) along with 24 prey

proteins interacting with all three versions of HA-tagged CDPK1 (Fig. 5b, Supplementary Table 5). Ribosomal proteins and chaperones/chaperonins involved in protein folding were well represented in this interactome analysis. We further analyzed those proteins using a network GO term analysis for molecular function ($p < 0.05$) and found that the terms structural constituents of the ribosome along with unfolded protein binding, GTP binding and GTPase activity were over-represented (Fig. 5c). We also tried to identify candidate targets for the kinase using a strategy described by Lebska et al[34]. where radioactive-immunocomplex based kinase assay was performed from immunoprecipitates of CDPK1-HA. Five bands, corresponding to 75 kDa, 55 kDa, 35 kDa, 25 kDa, and 20 kDa were detected preferentially with the CDPK1-HA version (Fig. 2a) and gel fragments retrieved from non-radioactive immunocomplex ran in parallel were studied by LC-MS/MS. This led to the identification of 206 proteins with at least two unique peptides identified with

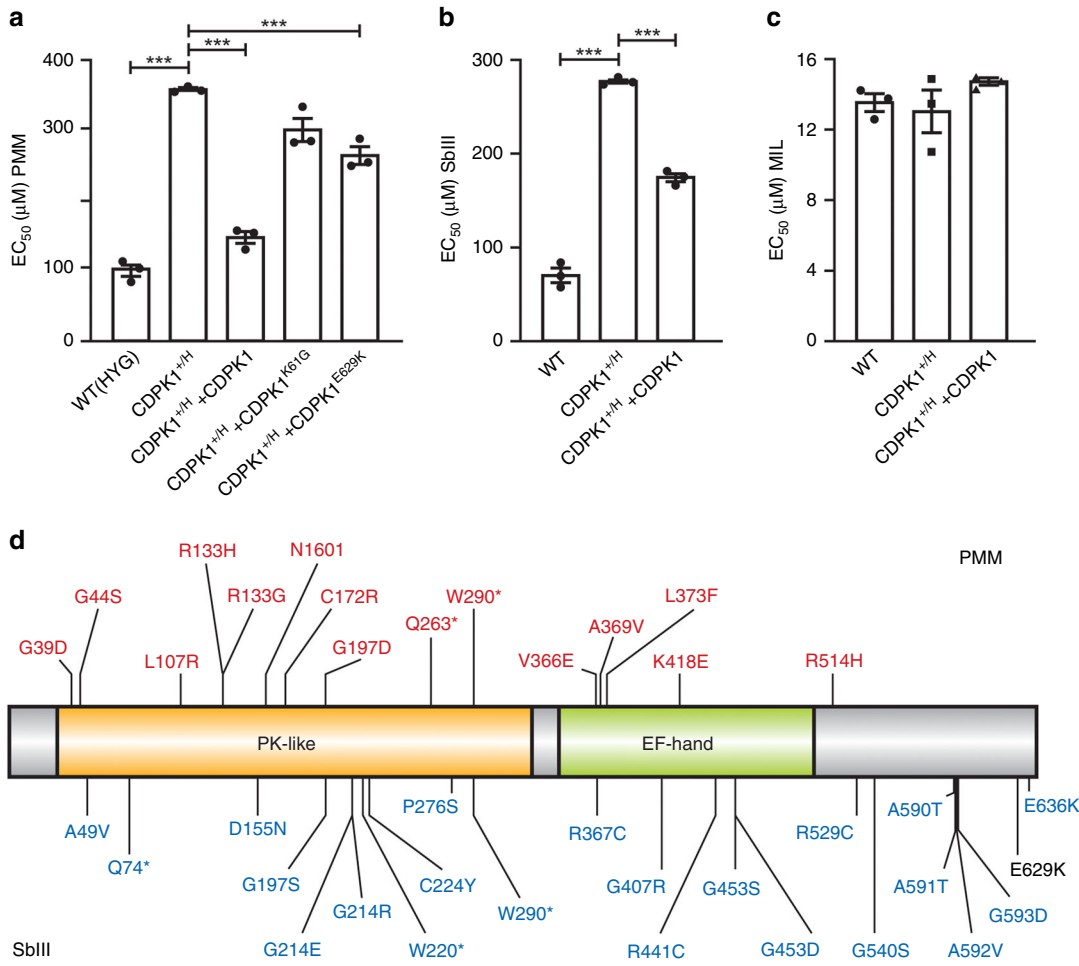

**Fig. 4 CDPK1 is linked to paromomycin and antimonial resistance. a** Dose responsiveness to PMM of the single knockout line $CDPK1^{+/H}$ complemented with an episomal rescue coding for the wild-type, K61G or E629K versions of CDPK1. Superscript $H$ and $+$ refer to the $HYG$-inactivated and wild-type alleles, respectively. Data are mean ± SEM for $n = 3$ biological replicates. Statistical analyses were performed using unpaired two-tailed $t$-tests.***$P < 0.001$; **$P < 0.01$. Dose responsiveness against SbIII (**b**) and miltefosine (**c**) for the single knockout line $CDPK^{+/H}$ complemented or not with an episomal rescue coding for the wild-type protein. Data are mean ± SEM for $n = 3$ biological replicates. Statistical analyses were performed using unpaired two-tailed $t$-tests. ***$P < 0.001$. **d** Mutations in CDPK1 detected in PMM (red) and SbIII (blue) resistant mutants. The mutation in black (E629K) was detected in one mutant selected by step wise exposure to SbIII in an independent study[32]. An asterisk represents a stop codon. Source data are provided as a Source Data file.

≥95% confidence and false discovery rate (FDR) <1% (Supplementary Data 4). These candidates were grouped using GO-SLIM terms into biological process (BP), cellular compartment (CC), and molecular functions (MF) (Fig. 6a). The enriched categories included proteins linked to ribosomal function, translation, and protein folding (Fig. 6a). Interestingly, out of the 206 proteins 40 were also identified as phosphorylated by a phosphoproteome analysis of *L. donovani* cells[35]. Among those 40 proteins, 9 proteins are linked to translation (Fig. 6b). The 206 proteins identified were analyzed by kinasephos2.0 (ref. [36]) for prediction of candidate AMPK/CaMK phosphorylation and 16 proteins, including two ribosomal proteins, were predicted with high confidence (Support Vector Machine value, SVM > 0.8) to contain this motif (Supplementary Table 6).

We tested the specificity of the proposed interactions by reciprocal IP by co-transfecting a CDPK1-TY1 version with 8 different HA-tagged proteins (Supplementary Fig. 9a), chosen from the lists of putative CDPK1 partners (Supplementary Table 5) or substrates (Supplementary Data 4). We immunoprecipitated individual co-transfections with HA and, under the conditions tested, the immunoprecipitates of two ribosomal proteins (L23a and L28), a P25-alpha, and ARM56 detected

CDPK1 by reacting to an anti-TY1 antibody (Supplementary Fig. 9a, b). We further focused on ribosomal proteins L23a and L28, as translation proteins are prevalent in the PMM Mut-seq mutants and that L23a has been associated with resistance to antimony in *Leishmania* and cross-resistance to PMM[37]. Constructs coding for HA-tagged CDPK1 and L23a and L28 were co-transfected, co-immunoprecipitated and we carried out an in vitro kinase assay. Under the conditions tested L23a but not L28 was found to be phosphorylated (Supplementary Fig. 9c). The phosphorylation of L23a by CDPK1 was competed by a peptide known to be phosphorylated by CDPK1 (Fig. 6c and Supplementary Fig. 9d). The K61G version of CDPK1-HA was also co-transfected with a L23a-HA expressing construct, co-immunoprecipitated and an in vitro kinase assay showed that the phosphorylation of L23a was impaired (Supplementary Fig. 9e).

Our Mut-Seq analysis and CDPK1 interactome and putative target analysis suggest that the translation machinery could be altered in the PMM mutants and that CDPK1 might act as a modulator. Incorporation of $S^{35}$ methionine was higher in the PMM25 mutant compared to wild-type cells and this was reversed by the episomal transfection of $CDPK1$ into PMM25 (Fig. 6d). Two-fold higher $^{35}S$ incorporation was also observed in

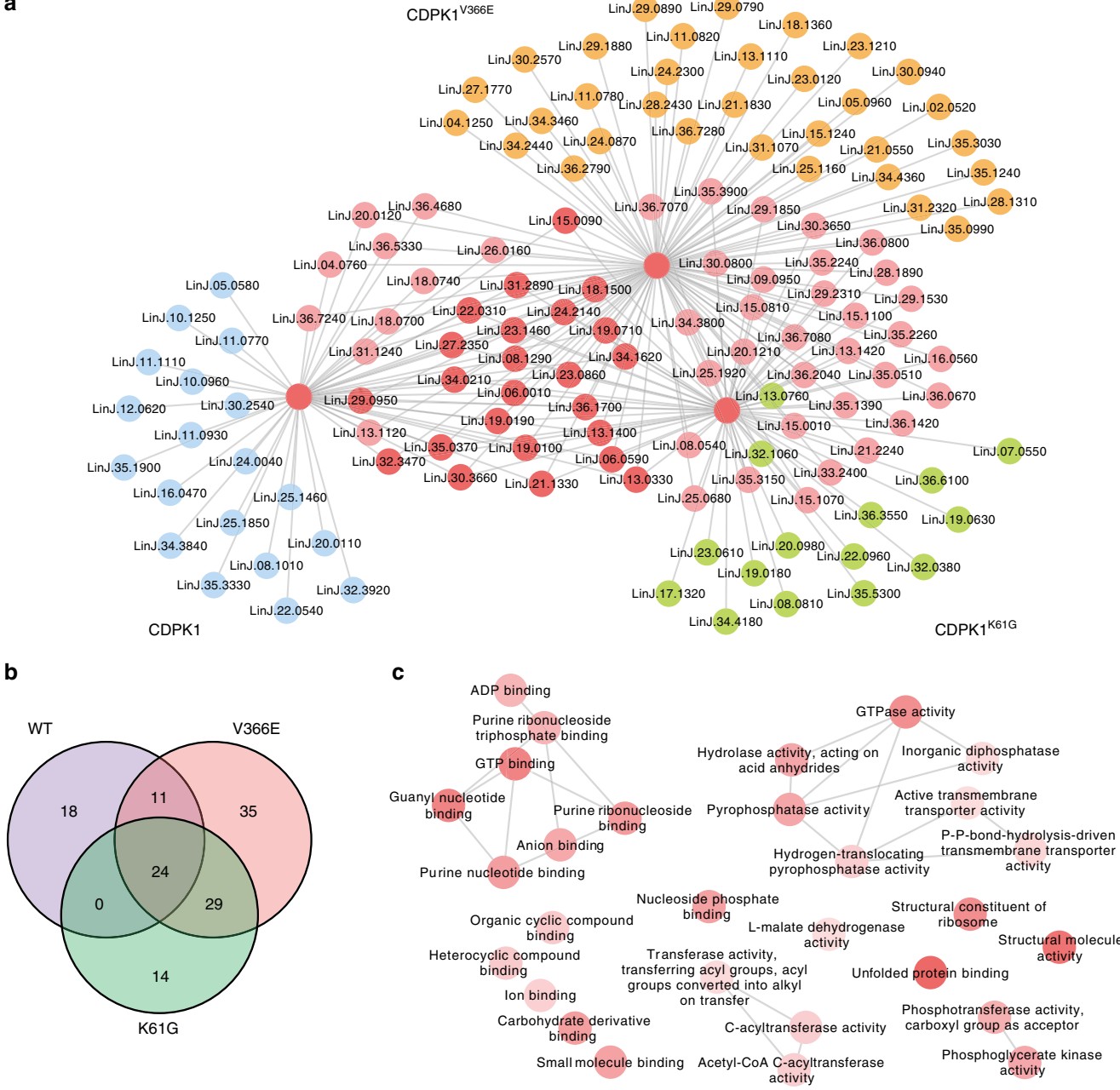

**Fig. 5 CDPK1 interactome and candidate targets. a** Nodes inferring specific gene-IDs as retrieved from SAINT analysis (AvP > 0.8) of IP data for CDPK1-HA, CDPK1[K61G]-HA and CDPK1[V366E]-HA. Visualization of predicted interacting partners by Cytoscape. Independent IP experiments, carried out using HA-DHFR-TS and HA-PTR1 expressed episomally in *L. infantum*, were treated as controls. Test and control experiments were set in three biological replicates. SAINT analysis was performed using total spectral counts for peptides identified with <1% FDR. The networks were merged using DyNet analyzer. Unique nodes are presented in orange, green and blue respectively for CDPK1[V366E]-HA, CDPK1[K61G]-HA and CDPK1-HA. Common nodes are presented in shades of red depending on the level of overlap. An enlarged version of the three interactomes is shown in Supplementary Fig. 8. **b** Number of overlapped proteins identified by immunoprecipitation of the three CDPK1-HA versions is shown by Venn diagram. Lists of unique proteins associated with CDPK1 and proteins present in all samples are found in Supplementary Data 3. **c** Cytoscape visualization of network depicting significantly (*P* < 0.05) over-represented GO-molecular function terms for the gene list generated by SAINT analysis of possible interactome for CDPK1. The network file was generated using REVIGO following analysis of the gene list for GO-enrichment using existing tools in TriTrypDB. Nodes represent specific GO-MF terms and intensity of shades signify extent of enrichment for the term.

the *CDPK1[+/H]* line in comparison to wild-type cells and this was reversed by the episomal add-back of *CDPK1* (Fig. 6d). This putative alteration in translation mediated by CDPK1 was further corroborated by polysome profiling where polyribosomes were found to be more abundant in the *CDPK1[+/H]* line (Polysomes/Monosomes ratio: 1.375) in comparison to the wild-type control or add back line (P/M ratio: 1.04) (Fig. 6e).

## Discussion

Whole genome gain- and loss of function associated with drug selections and NGS are well poised for studying resistance involving gene overexpression[7,38] or gene deletion/inactivation[8,39] but are less suited for point mutations. The Mut-Seq strategy has recently been established, facilitating studies on the role of point mutations in resistance. This technique is more suitable for

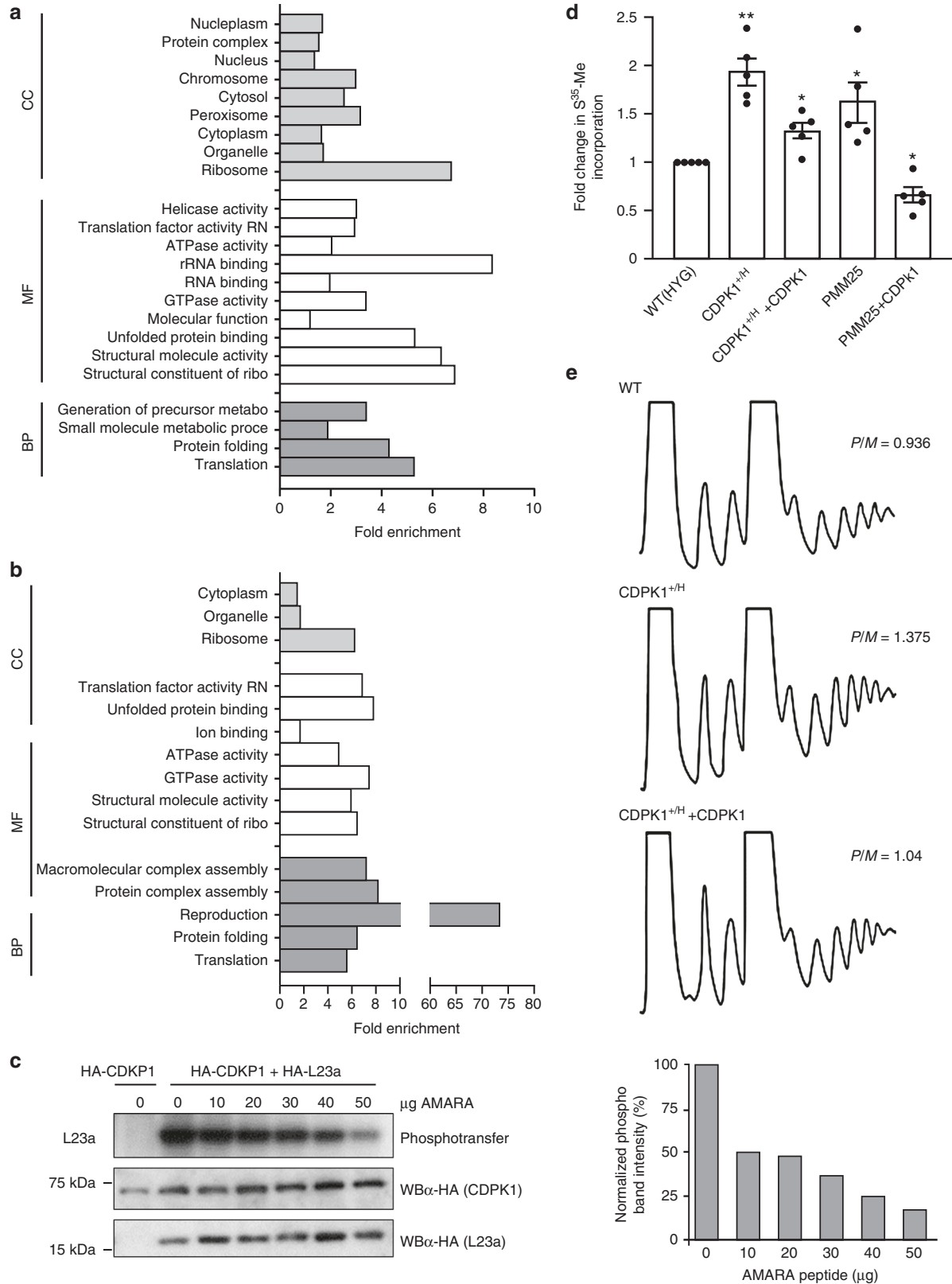

organisms that have a life stage with an haploid genome but this is now used for organisms with diploid or aneuploid genomes[16]. *Leishmania* possess a diploid genome but with considerable aneuploidy and even mosaic aneuploidy[40]. Our EMS treatment led to ~8 non-synonymous SNVs/Mb of diploid genome which is higher compared to other screens in other systems at ~1 non-

synonymous SNV/Mb[19], a mutation rate observed for the ENU mutagenesis. Lower frequency (~0.3 SNVs/Mb) was observed with HMPA or MMS screens. Of the four mutagens used, EMS (responsible for C to T and G to A transition) was found to be the most potent, possibly because a higher toxic concentration of the mutagen (4X EC$_{50}$) could be used and because of the high GC

**Fig. 6 CDPK1 modulates translation efficiency.** Functional GO-annotation of the 206 proteins identified by LC-MS/MS from phosphorylated bands derived from SDS–PAGE of in vitro IP kinase reaction of CDPK1-HA (**a**); Functional GO-annotation of the 40 proteins (out of the 206) reported to be phosphorylated by Tsigankov et al. (2013) (**b**). Proteins were grouped functionally based on GO-slim terms. Fold enrichment of particular GO-slim terms describes ratio of representation of the terms in the identified pool with that in the genomic repertoire. Enriched GO-terms for biological process (BP), molecular function (MF), and cellular component (CC) are shown. **c** A competitive kinase assay was performed in the presence of increasing concentrations of AMARA peptide, a CDPK1 substrate (Supplementary Fig. 9d), using immunoprecipitated lysates (anti-HA antibody) prepared from *L. infantum* expressing HA-CDPK1 alone or in combination with HA-L23a. The left panel shows the phosphotransfer reaction (top panel) and the amount of immunoprecipitated HA-CDPK1 (middle panel) and HA-L23a (bottom panel) involved in each reaction, as observed by western blotting (WB) of the immunoprecipitated complex using mouse anti-HA (α-HA) IgG. The normalized phosphotransfer signal intensities for the different AMARA peptide concentrations are shown on the right panel. **d** Active translation was monitored by $S^{35}$-Methionine incorporation assay in log-phase promastigotes from the strains mentioned. The fold change of $S^{35}$-Methionine incorporation in TCA precipitated protein fractions were compared with wild-type cells. Superscript *H* and +refer to the *HYG*-inactivated and wild-type alleles, respectively. Data are mean ± SEM for n = 5 biologically independent experiments. Statistical analyses were performed using paired two-tailed *t*-tests. \*\*$P < 0.01$; \*$P < 0.05$. **e** Polysome profile of wild-type, $CDPK1^{+/H}$ and $CDPK1^{+/H} + CDPK1$ parasites in exponential growth phase. The polysome/monosome ratio was determined by measuring the area of the polysome peaks and of the 80S monosome peak using the ImageJ software (developed and maintained by the National Institutes of Health, Bethesda, MD). Shown are representative profiles from two independent experiments. Source data are provided as a Source Data file.

content (61.1%) of *Leishmania*. The use of higher concentrations of ENU, MMS or HMPA was toxic with no colonies growing on plates with selective drugs. Since resistance levels are similar with the different mutagens (Fig. 1a), a condition with less SNVs could theoretically facilitate the discovery of phenotypic SNVs. However, HMPA and MMS that were associated with less SNVs failed to select for PMM or MIL mutants, respectively. Thus mutagens such as EMS or ENU associated with more SNVs are useful since they allow a sufficient number of mutants but also strengthen our approach of studying converging mutated candidates.

Around 500 SNVs were observed with EMS mutagenized cells selected with drugs, which required a hierarchy of the mutations. Homozygous mutations were first prioritized. While rare, they were observed in this diploid organism. These may have originated from loss of heterozygocity, a well described phenomenon in *Leishmania*[41]. For heterozygous mutations, we focused on recurrent mutations in multiple independent mutants. One MIL mutant had a homozygous mutation in MT and 10 additional mutants had heterozygous mutations. This is an example of loss of function since some mutations were non-sense and numerous reports have associated a loss of MIL transport when MT is mutated[25,42]. This study highlighted also genes involved in lipid metabolism and associated with MIL resistance. Our preferred route for studying the role of several genes consisted in the episomal transfection of wild-type genes in a mutant. As there were many mutations in each mutant we had expected that many mutations are involved in the phenotype and that a given gene will only partly restore susceptibility; an expectation indeed observed (Table 1). For a fatty acid elongase (LinJ.14.0790) and the MIL transporter MT we confirmed their role in resistance both by episomal transfection and by DNA editing. The orthologue of LinJ.14.0790 in *L. major* groups with the *T. brucei* fatty acid elongases TbELO1–3, a clustering that suggests a likely role in elongating saturated fatty acid chains[43]. The product of *LinJ.14.0790* indeed harbors a GNS1/SUR4 family domain whose members are involved in long-chain fatty acid elongation systems that produce the 26-carbon precursors for ceramide and sphingolipid synthesis. Interestingly, depletion of sphingolipid was shown to lead to a drastic alteration in sterol metabolism in *L. major* associated with a 3-fold reduction in sensitivity to MIL[44]. It is also noteworthy that mutations in LinJ.14.0790 were observed in conventionally-raised MIL-resistant mutants. This further reinforces the linkage between mode of action of MIL and lipid metabolism[44,45]. Several other genes may further be associated with MIL susceptibility and await further experimental confirmation. Of the ones with annotated putative function 24 can be associated with lipid metabolism (Supplementary Data 1).

Resistance to MIL was shown to correlate with the overexpression of a P-glycoprotein in a multi-drug resistant *L. tropica*[46] and ABC proteins can thus affect susceptibility to MIL. Surprisingly here we found that a mutation in the 3′UTR of the gene coding for ABCC8 (ref. [23]) alters gene expression and impact MIL susceptibility by a mechanism that remains to be studied.

PMM was recently introduced for treating *Leishmania* infections. As an aminoglycoside, a likely target would be the translation machinery. While initial studies of PMM-resistant parasites indicated reduced uptake of the drug[47,48], recent proteomic efforts supported the idea that PMM affects the *Leishmania* translational machinery[49]. This is further reinforced by recent structural analysis of the *Leishmania* ribosome complex with PMM[50]. Consistent with the role of PMM in translation we found several heterozygous mutations in proteins involved in translation in independent mutants, some of which were found to contribute to PMM susceptibility (Table 1).

A marker of PMM resistance highlighted was CDPK1. This protein kinase appears essential for parasite growth. Alteration in this kinase (through mutation or inactivation of one allele) leads to resistance and the kinase activity of CDPK1 appears important for this phenotype. CDPK1 likely controls some of the translational processes, as homozygous mutation or inactivation of one *CDPK1* allele increases translation as measured by $S^{35}$ incorporation and polysome profiling. This is consistent with the increase in ribosomal protein expression, and possibly of translation rates, previously described in PMM-resistant *L. donovani*[49]. The *Leishmania* CDPK family is small; with two proteins that cluster with their *Trypanosoma* orthologues together with the CDPK6 of Apicomplexa. A decrease in the activity of the essential malaria CDPK1 was associated with CDPK6 overexpression and was suggested as a compensatory mechanism[51] and while the *Toxoplasma* CDPK6 is not essential it was associated with parasite fitness and infectivity[52]. It remains to be seen whether the Apicomplexa proteins and the *Leishmania* CDPK1 share activities. The *Leishmania* orthologue differs in containing a long flexible C-terminal stretch (Supplementary Fig. 4a). Immunoprecipitation of CDPK1 has highlighted a variety of putative partners and the C-terminal extension may contribute to those interactions. Interestingly, a number of potential interacting proteins were validated by reciprocal immunoprecipitation and some are involved in translation (ribosomal proteins L23a and L28). Future work will help in determining all the legitimate CDPK1-substrate interactions. The ribosomal protein L23a not only interacted with CDPK1 but it was also phosphorylated by it, further reinforcing the role of the *Leishmania* CDPK1 in translation. Mutations in CDPK1 impacted its kinase activity and

consequently the level of L23a phosphorylation. Dephosphorylation of ribosomal proteins have been shown to increase the activity of ribosomes in translation in prokaryotes[53]. In humans, the phosphorylation status of ribosomal proteins influences the translation of distinct subpools of mRNAs[54]. The link between increased translation rates and CDPK1 mutations remains to be established, but negative charges from phosphoryl groups could for example hamper with ribosomal subunit association, influence the binding of mRNA to ribosomes, modulate binding of initiation factors to ribosomes, or affect the peptidyl transferase reaction. Since PMM can induce ribosomal infidelity and promote mis-incorporation of amino acids, hence culminating into protein misfolding[55], modulation of chaperone action along with translational regulation could be a determinant in the PMM susceptibility of the parasite. Indeed, CDPK1 potentially interacts with chaperonins, ribosome-associated chaperones, and proteases.

One unanticipated observation was the role of CDPK1 in SbIII cross-resistance but this is consistent with (i) our initial finding that CDPK1 was mutated in a cell line selected for resistance to SbIII[32] and (ii) of a recent study also linking a single mutation in CDPK1 and resistance to SbIII[56]. Interestingly, both the interactome study and the putative CDPK1 phosphorylation target identification highlighted ARM56 (LinJ.34.0210), a known antimony resistance marker[57]. This interaction was further confirmed by reciprocal IP. Further work could reveal if there are any interactions between ARM56 and CDPK1 affecting SbIII susceptibility. It is possible that CDPK1 confers resistance to PMM and SbIII by distinct mechanisms by phosphorylating different proteins.

In this study, we have combined chemical mutagenesis and NGS in *Leishmania* and the capacity to sequence several clones and to look for biological recurrence of mutations facilitated the discovery of genes associated with a phenotype. We highlighted the role of genes involved in lipid metabolism in the mode of action of MIL and a mechanism of PMM resistance targeting an essential kinase that seems to regulate the processes of protein translation and protein folding. The technique is relatively straightforward and could be applied to different *Leishmania* species to fetch general resistance mechanisms but also for testing whether species-specific mechanisms also exist. This technique of Mut-Seq can be successfully used with diploid and aneuploid organisms and thus should be widely applicable.

## Methods

**Parasites and macrophage culture.** *L. infantum* JPCM5 (MCAN/ES/98/LLM-877) and *L. infantum-263* (MHOM/MA/67/ITMAP-263) parasites were maintained as promastigotes at 25 °C in SDM-79 or M199 medium supplemented with 10% (vol/vol) heat-inactivated FBS and 5 μg/mL hemin. Cell growth was monitored by measuring the absorbance at 600 nm in a multiwell scanning spectrophotometer (Multiskan, Thermo Scientific, Waltham, MA, USA). EC$_{50}$ values were determined by measuring the absorbance at 600 nm of culture aliquots grown in the presence of various concentrations of drugs in a multiwell scanning spectrophotometer. The cell line THP-1 (ATCC, cat no. TIB-202) was cultured and differentiated by incubation for 2 days in medium containing 20 ng of phorbolmyristate acetate (PMA)/ml. PMA-differentiated THP-1 macrophages were infected with stationary-phase parasites at a ratio of 30:1, for 3 h at 37 °C in a 5% CO$_2$ atmosphere. Cells were maintained in drug-free medium for 48 h after which infected cells were either left untreated or treated with PMM for 96 h at 37 °C. The number of infecting amastigotes per macrophages was determined by Giemsa staining of the infected cells followed by microscopic observation. The parasitic index (P$_{Idx}$) was calculated as the percentage of infected cells multiplied by the mean number of parasites per cell.

**Chemical mutagenesis.** A source culture for chemical mutagenesis was initiated with a freshly picked *L. infantum* clone inoculated in SDM-79 medium supplemented with 10% (vol/vol) heat-inactivated FBS and 5 μg/mL hemin. In all, $5 \times 10^7$ early log phase *L. infantum* promastigotes were transferred to each of a series of culture flasks with SDM-79 medium supplemented with 10% (vol/vol) heat-inactivated FBS and 5 μg/mL hemin. EMS, ENU, MMS, or HMPA were added to

these flasks to a final concentration of 40 mM, 4 mM, 0.1 mM, and 100 mM respectively and incubated for 6 h at 25 °C. Cells were washed with 1X HEPES-NaCl twice and regenerated in fresh medium for 24 h to 36 h depending on the level of recovery post-mutagenesis. In all, $5 \times 10^6$ regenerated cells were plated on 1% SDM-79 agar supplemented with 10% (vol/vol) heat-inactivated FBS and 5 μg/mL hemin containing PMM or MIL (Supplementary Table 1). Plates were incubated at 25 °C for 10 days for emergence of resistant clones. Each clone was revived in 5 ml of SDM-79 and analyzed for drug responsiveness and genomic DNA preparation at passage one. Medium and wash buffers containing mutagens were neutralized with excess NaOH. The source clone was maintained in parallel with the mutants. For drug susceptibility assays, the drug EC50s of the mutants were compared to the one of the source clone.

**Whole-genome sequencing and analysis.** Genomic DNA was prepared from a mid-log phase clonal culture of *L. infantum*. Paired-ends sequencing libraries were prepared with the Nextera DNA sample prep kit and sequenced on an Illumina HiSeq platform 1000 with 101-nucleotide paired-ends reads. An average genome coverage of over 50-fold was aimed for the mutants (Supplementary Fig. 1a). Sequence reads were aligned to the *L. infantum* JPCM5 genome using the software bwa-mem[58]. The maximum number of mismatches was 4, the seed length was 32 and 2 mismatches were allowed within the seed. Read duplicates were marked using Picard (http://broadinstitute.github.io/picard) and we applied GATK for SNVs and indels discovery[59]. PCR amplification and conventional DNA sequencing confirmed the SNVs revealed by NGS. Copy numbers variations were derived from read depth coverage by comparing the coverage of uniquely mapped reads between wild-type and mutagenized parasites along small non-overlapping genomic windows (5 kb) for the 36 chromosomes (normalized to the total number of uniquely mapped reads for each strain). Several python and bash scripts were created to further analyze the data.

Calling for point mutations using GATK predicted a large number of SNVs, particularly for EMS-mutagenized clones. To avoid false calling for SNVs, we filtered the sequences with the sequence of our wild-type *L. infantum* strain to the JPCM5 reference strain. This was followed by validation of a training set of 60 intragenic SNVs by Sanger sequencing of targeted PCR amplified regions harboring predicted SNVs. On the basis of the validation, a cut-off quality score of 150 was found to be a strong predictor for calling true SNVs from NGS data.

**DNA constructs and transfection.** The genes of *L. infantum* were amplified from genomic DNA using compatible primer pairs and PCR fragments were ligated into pGEM T-easy (Promega, Madison, WI, USA) for confirming the quality of the insert by standard sequencing, and then cloned in the *Leishmania* expression vector pSP72αZeoα, pSP72αPuroα or pSP72αHYGα. A total of 20 μg of plasmid DNA for episomal expression, either the empty vector (mock) or carrying the genes of interest, were transfected into *Leishmania* promastigotes by electroporation with Gene Pulser Xcell (Biorad) at 450 V, 500 μF and time constant range in 4–6 millisec[60]. For preparing knock out construct, either 300 bp or 600 bp from 5′ and 3′ UTR of the target genes were amplified along with the selective marker gene by a standard PCR-fusion based strategy. The fusion fragment was purified and cloned into pGEM T-easy to prepare pGEM T-easy-LinJ33.1810KOpuro and pGEM T-easy-LinJ33.1810KOhygro plasmids and the quality of insert was confirmed by standard sequencing. The cassettes were generated by either digesting the plasmid with NotI or by PCR with terminal primers. 10 μg of the purified cassettes were transfected in log-phase *L. infantum* promastigotes and selected with optimized selection pressure for respective markers (zeocine: 400 μg/ml; puromycin: 100 μg/ml; hygromycin: 400 μg/ml and blasticidine: 100 μg/ml).

**CRISPR-Cas9 based genome editing.** For CRISPR-Cas9 based genome editing, the ORF coding for the CRISPR associated protein 9 (Cas9) nuclease of *S. pyogenes* was amplified from the CMV-CAS9-2A-GFP commercial vector (Sigma-Aldrich) and cloned in the *Leishmania* expression vector pSP72αHygroα and transfected in the parasite to generate *L. infantum* JPCM5 pSP72αHygroαCas9 lines[61]. Two guide RNAs targeting the 5′ and 3′ UTR of CDPK1 were designed targeting the following sequences: gRNACDPK1-5: AACGGATTAAAGGTCAGATAC and gRNACDPK1-3: CGTGAAACCCCGTTGTCCTC (IDT). A homology repair template (HRT) was amplified from genomic DNA of mutant or WT cells using primers with substitution in the gRNA targeting PAM sites- *CDPK1repair-F*: GTGCGGGTTGTG CACTTCACTCT GCACTGTCTTGTATCTGACCTTTAATCGTT and *CDPK 1repair-R*: GTGCGGGTTG TGCACTTCACTCTGCACTGTCTTGTATCTGA CC TTTAATCGTT. One heterologous selection cassette was prepared by generating construct for homologous recombination based allele replacement targeting PTR1 gene (PTR1KO-Zeo). In all, 5 μl of 0.1 nmol/μl of gRNA (CrRNA) was hybridized with 5 μl of equimolar tracrRNA (IDT) after resuspension with IDTE Buffer to a final concentration of 200 μM. The two RNA oligonucleotides were mixed to a final duplex concentration of 100 μM, heated at 95 °C for 5 min and allowed to cool down to room temperature (15–25 °C)[61]. 8 μg of repair template, 5 μl of each CrRNA-tracrRNA hybrid and 10 μg PTR1KO-Zeo was transfected simultaneously using Amaxa-Nucleofector™ transfection kit (Lonza). The selection was proceeded with zeomycin (400 μg/ml) and allelic substitution was confirmed by PCR amplification of target gene followed by standard sequencing.

**Immunoprecipitation and samples for mass spectrometry**. Immunoprecipitation was done using Pierce™ HA-Tag Magnetic IP/Co-IP Kit according to manufacturer's protocol. Briefly, pellets derived from mid-log phase cells were washed twice with 1XPBS and resuspended in the lysis buffer supplemented with Halt™ Protease Inhibitor Cocktail and Halt™ Phosphatase Inhibitor Cocktail (Thermo Scientific). Lysis was completed by 20–30 strokes of a Dounce homogenizer while on ice. Cell debris and insoluble material were separated by 30 min centrifugation at 10,000×g at 4 °C. Clear supernatants were incubated with anti-HA magnetic beads at 4 °C for 4 h on a gentle rotator. Beads were then washed thrice with lysis buffer (30 sec each) and thrice with 50 mM ammonium bicarbonate by gentle agitation. Those were suspended in 25 μl 50 mM ammonium bicarbonate and trypsin (1 μg) was added and sample was incubated overnight at 37 °C. Trypsin reaction was stopped by acidification with 3% acetonitrile-1% TFA-0.5% acetic acid. Beads were removed and the peptides were purified on stage tip (C18), vacuum dried and solubilized into 10 μl of 0.1% formic acid before MS injection.

**LC MS/MS analysis**. Peptide samples were separated by online reversed-phase nanoscale capillary liquid chromatography (nanoLC) and analyzed by electrospray mass spectrometry (ES MS/MS). The experiments were performed with a Ekspert NanoLC425 (Eksigent) coupled to a 5600 + mass spectrometer (Sciex, Framingham, MA, USA) equipped with a nanoelectrospray ion source. Peptide separation took place on a picofrit column (Reprosil 3 u, 120 A C18, 15 cm × 0.075 mm internal diameter, New Objective, Woburn, MA). Peptides were eluted with a linear gradient from 5–35% solvent B (acetonitrile, 0.1% formic acid) in 35 min, at 300 nL/min. Mass spectra were acquired using a data dependent acquisition mode using Analyst software version 1.7. Each full scan mass spectrum (400–1250 m/z) was followed by collision-induced dissociation of the twenty most intense ions. Dynamic exclusion was set for a period of 20 s and a tolerance of 100 ppm.

**Data processing**. MGF peak list files were created using Protein Pilot version 4.5 software (Sciex). MGF sample files were then analyzed using Mascot (Matrix Science, London, UK; version 2.5.1). Mascot was set up to search the LeishInfant_LinfantumAnnotatedProteins_TriTrypDB-4.0_20120202 database (8241 entries) assuming the digestion enzyme trypsin. Mascot was searched with a fragment ion mass tolerance of 0.100 Da and a parent ion tolerance of 0.100 Da. Deamidation of asparagine and glutamine and oxidation of methionine were specified in Mascot as variable modifications. Scaffold (version Scaffold_4.7.1, Proteome Software Inc., Portland, OR) was used to validate MS/MS based peptide and protein identifications. Peptide identifications were accepted if they could be established at greater than 51.0% probability to achieve an FDR less than 1.0% by the Scaffold Local FDR algorithm. Protein identifications were accepted if they could be established at greater than 99.0% probability to achieve an FDR less than 1.0% and contained at least 2 identified peptides. Protein probabilities were assigned by the Protein Prophet algorithm[62]. For immunoprecipitation of CDPK1-HA, this led to the identification of 298 proteins with atleast 2 unique peptides (≥95%) in any one of the biological replicates with <1%FDR. Bait recovery of the precipitate yielded average of 301.33 for spectral count with average coverage of 72.33. Similar analysis performed with CDPK1$^{K61G}$-HA and CDPK1$^{V366E}$-HA resulted bait recovery of average spectral count of 72.33 and 80.33 with coverage of 57 and 60% respectively.

**Immunocomplex protein kinase assay**. To determine kinase activity, immunoprecipitation was performed with 10⁹ mid-log phase cells. The immuneprecipitated magnetic beads were washed thrice with lysis buffer and then resuspended in kinase assay buffer (NEB). A portion of the resuspended beads were analyzed by Western blotting with mouse anti-HA IgG (Santa Cruz Biotechnologies, cat no. Sc-7392; 1:3000 dilution). Unprocessed blots can be found in the Source Data file.

For identification of probable substrates for CDPK, the assay was analyzed by SDS–PAGE (12%). A replicate set of reaction was performed with non-radioactive ATP. Both sets were ran on SDS–PAGE and signals were detected by autoradiography. Specific bands parallel to the signal in the γ-P³²-ATP containing reaction were cut from the non-radioactive reaction set and further analyzed for LC-MS/MS analysis.

For validation of candidate target proteins, HA-tagged versions of putative targets and of CDPK1 were co-expressed in promastigotes. Immunoprecipitation was performed using anti-HA antibody (see above), which was followed by kinase assay. Phosphotransfer to low molecular ribosomal proteins was analyzed after reaction in 15% SDS–PAGE. For determining phosphotransfer specificity, the kinase reaction was performed in the presence of 1 μg AMARA peptide, following which samples were spotted on P81 phosphocellulose paper, washed twice with 0.5% phosphoric acid and analyzed by scintillation counting.

**Reciprocal immunoprecipitation**. N-terminal HA-tagged version of putative partners and 2 × -Ty1-tagged CDPK1 were co-expressed in CDPK1$^{+/H}$ promastigotes. Immunoprecipitation was performed from 10⁹ cells using the Pierce™ HA-tag Magnetic IP/Co-IP Kit according to the manufacturer's instructions. Immunoprecipitates were washed twice with 1 × PBST and resuspended in Laemmli

buffer. Western blot was performed with the immunoprecipitates using rabbit anti-Ty1 antibody (GenScript, cat no. A01004; 1:500 dilution).

**Bioinformatic analysis**. To identify proteins that interacted specifically with CDPK1 and reduce false interactions, two independent negative controls were performed (Immunoprecipitation performed targeting LiDHFR-TS and LiPTR1 using identical strategy, three biological replicates for CDPK1, LiDHFR-TS and LiPTR1).

Using the negative controls, significance analysis of interactome (SAINT) was performed, based on normalized spectral count comparison[33]. The probability scores of the bait and prey proteins were calculated as the average of the probabilities in individual replicates (AvgP). Proteins with AvgP ≥ 0.8 were considered as likely interactors.

Gene ontology was performed using TriTrypDB resources. Interactors were categorized into cellular compartment, molecular function, and biological process terms. Only enriched GO terms with a p value <0.05 were selected. CDPK1-specific phosphorylation sites were predicted using KinasePhos2.0 (ref. [36]) for AMPK/CaMK phosphorylation motifs with default prediction threshold. The identified proteins were further screened with cuff-offs for SVM values 0.7 and 0.8 to predict moderate to high-confidence candidate substrates.

Phylogenetic analysis was performed using MEGA6 (ref. [63]). Heat maps were prepared using Heatmapper[64].

**Western blot**. SDS–PAGE was performed on 10 or 12% acrylamide gels according to standard procedures. The BioRad Kaleidoscope ladder was used (Life Technologies, Carlsbad, CA, USA). Protein expression was detected using the Immobilon western chemiluminescence kit (Millipore, Billerica, MA, USA). Antibodies used are as follows: mouse anti-HA IgG (SantaCruz Biotechnologies, cat no. Sc-7392; 1:3000 dilution), mouse anti-Tubulin IgG (Milipore, cat no. T6199; 1:15,000 dilution), mouse anti-HSP70 (Abcam, cat no. ab2787; 1:3000 dilution), mouse anti-HSP60 (Assay Design, cat no. SPA-807–488;1:1000 dilution), mouse anti-BIP (a kind gift from Jay Bangs; 1:5000 dilution) and anti-mouse-HRP conjugated IgG (Thermo Scientific, cat no. 31430; 1:10,000 dilution).

**Polysome Profiling Analysis from sucrose gradient**. A total of $3 \times 10^9$ *L. infantum* early-log phase promastigotes were incubated with 100 μg/ml cycloheximide (Sigma) for 10 min, washed with cycloheximide-containing PBS buffer and lysed with a Dounce homogenizer in lysis buffer [10 mMTris-HCl pH 7.4, 150 mMNaCl, 10 mM MgCl2, 1 mM DTT, 0.5% IGEPAL, 100 μg/ml cycloheximide, 100 U/ml RNAseOUT (Amersham), 1 mM PMSF, 15 μl/ml of protease inhibitor cocktail (Sigma)]. *Leishmania* lysates were pelleted by centrifugation at 12,000 rpm for 15 min at 4 °C, and the supernatant (40 OD 260 nm units) was layered on top of a 15–45% linear sucrose gradient (10 ml) in gradient buffer (50 mM Tris-HCl pH 7.4, 50 mM KCl, 10 mM MgCl2, 1 mM DTT, 100 U/ml RNaseOUT)[65]. Ribosomal subunits (40S and 60S), monosomes (80S) and polyribosomes were sedimented by centrifugation in a Beckman SW40 Ti rotor at 35,000 rpm for 2.15 h at 4 °C and fractions were collected using an ISCO Density Gradient Fractionation System under constant monitoring of the absorbance at 254 nm.

**S³⁵ methionine incorporation**. $10 \times 10^6$ early log-phase *L. infantum* promastigotes were washed twice with 1XPBS and transferred to 1 ml methionine free RPMI-1640 (Gibco, Thermo Fisher Scientific) for 1 h at 37 °C (Gibco). In all, 2 μCi of S³⁵-methionine was added and the incubation was continued for 30 mins. Cells were pelleted and washed thrice with ice-cold PBS and lysed with 1X Lamelli buffer. In all, 10 μl of the lysate was spotted on Whatman paper and precipitated with 20% TCA, washed twice with ice-cold 95% ethanol followed by scintillation counting.

**Statistical analysis**. The statistical analysis were performed using two-tailed unpaired t-test with GraphPad Prism 5.01 software unless mentioned otherwise.

**Reporting summary**. Further information on research design is available in the Nature Research Reporting Summary linked to this article.

## Data availability

The dataset supporting the conclusions of this article is available in the Sequencing Read Archive (https://www.ncbi.nlm.nih.gov/sra) repository with accession numbers: MIL1 SRX4453621, MIL2 SRX4453626, MIL3 SRX4453627, MIL4 SRX4453656, MIL5 SRX4453657, MIL6 SRX4453654, MIL7 SRX4453655, MIL8 SRX4453652, MIL9 SRX4453653, MIL10 SRX4453650, MIL11 SRX4453622, MIL12 SRX4453623, MIL13 SRX4453624, MIL14 SRX4453625, MIL15 SRX4453618, MIL16 SRX4453619, PMM1 SRX4453651, PMM2 SRX4453633, PMM3 SRX4453632, PMM4 SRX4453631, PMM5 SRX4453630, PMM6 SRX4453637, PMM7 SRX4453636, PMM8 SRX4453635, PMM9 SRX4453634, PMM10 SRX4453658, PMM11 SRX4453659, PMM12 SRX4453639, PMM13 SRX4453647, PMM14 SRX4453648, PMM15 SRX4453649, PMM16 SRX4453642, PMM17 SRX4453643, PMM18 SRX4453644, PMM19 SRX4453645, PMM20 SRX4453640, PMM21 SRX4453638, PMM22 SRX4453646, PMM23

SRX4453641, PMM24 SRX4453629, PMM25 SRX4453628. Proteomics data supporting the findings of this study are available within the paper and in Supplementary Data 3 and 4. Other data sets and materials generated during the current study are available from the corresponding author on reasonable request. The list of PCR primers used in this study is provided as Supplementary Data 5.

The source data underlying Figs. 1a, 2a, b, 3b–d, 4a–c, 6c–e, Table 1, Supplementary Figs. 1a–d, 2a, c, 3a–c, 5a, c, d, 6a, b, 7d, 9b–e and Supplementary Table 4 are provided as a Source Data file.

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

## Acknowledgements

This work was supported by a Canadian Institutes of Health Research Foundation Grant to MO. M.O. holds a Canada Research Chair in Antimicrobial Resistance. We are thankful to the Proteomics core facility of the CHU de Quebec Research Center. We are thankful to Jay Bangs for providing the anti-BIP antibody.

## Author contributions

M.O., P.L., and B.P. designed and supervised the work. A.B., S.B., P.K.P., G.R., H.G., A.M.(Mestdagh) and A.M.(Mukherjee) performed the experiments. A.B. and P.L. performed the analyses and wrote the first draft of the manuscript. All authors interpreted the data and read, revised, and approved the final version of the manuscript.

## Competing interests

The authors declare no competing interests.
