## [Peer Review File · Nature Communications]

Reviewers' Comments:

Reviewer #1:

Remarks to the Author:

In this manuscript, Arijit Bhattacharya et al. address one relevant study concerning the use of chemical mutagenesis and next generation sequencing to facilitate the discovery of genes associated with drug resistance, using as a model the protozoan parasite *Leishmania*. Authors focused the study of resistance to antileishmanial drugs miltefosine and paromomycin with identification of known and new genes involved in drug resistance. New genes involved in lipid metabolism were associated with the mode of action of miltefosine, and a new protein kinase (CDPK1) was found crucial for paromomycin resistance.

The search for new drug resistant markers is relevant for these neglected tropical diseases. Specifically, leishmaniasis have a reduce arsenal of drugs that are toxic, expensive and with a high level of drug resistance/therapeutic failure.

In general, this is a relevant manuscript with interesting results obtained after the use of a well-designed, statistically supported and elegant methodology that will contribute significantly to the search of drug resistance markers to identify patients with drug resistance and to stablish new therapeutic strategies to control this parasitic disease.

However, there is some important issues in the manuscript that needs to be considered:

-Why authors do not have used the Mut-Seq strategy to identify drug resistant markers for Amphotericin B?. Considering that AmB is one of the most relevant antileishmanial drugs, authors must have some consideration to the search of drug resistant markers for AmB using this methodology. Additionally, few studies have success in the obtention of drug resistance to AmB; probably the strategy Mut-Seq could facilitate the acquisition of drug resistance and the selection for drug resistant markers. It is necessary that authors indicate in the manuscript some reference to this point (probably Discussion Section).

-Could the authors validated the most relevant identified drug resistant markers in clinical/experimental drug resistant *Leishmania* isolates?. Although, no clinical resistant isolates to PMM have been reported, probably by its low routine use in the field, authors could analyse in experimental PMM resistant lines the selected new markers of PMM resistance (specifically the protein kinase CDPK1).

-CDPK1 could be linked to MIL and/or AmB resistance?. Considering the involvement of this protein kinase in the PMM and SbIII resistance, it would be interesting that authors demonstrate the possible involvement of this kinase in the resistance to MIL and/or AmB in *Leishmania*.

-Other relevant question concerns about if drug resistant markers identified in *L. infantum* using the Mut-Seq strategy could be generalized for others *Leishmania* species, or on the contrary for each *Leishmania* species, different drug resistant markers could be obtained using Mut-Seq. Authors must indicate in the manuscript reference to this point.

-The EMS, as the most potent mutagen, could have similar effects in different *Leishmania* species, and consequently could determine different mutations and different drug resistant markers?.

Minor points:

-Table 1 – Title as stated in the present form is confusing considering that just indicate: in relevant MIL resistant mutants, but authors also refers to MIL and PMM.

-Table S4 and S5, can not be shown properly.

- Correct wild-type through all manuscript. Authors sometimes write wild-type with and without - .
- Fig. S7c, authors must specify the meaning of * in the mutants.
- Fig. S8 legend: authors refer to Fig 4a and not to Fig. 5a.
- Legend Table S9 refers to Table 3, however this Table was not present in the manuscript.
- Table 2 was not present in the manuscript.
- Pag 13, Results (line 9), authors refers to Fig. 5e that was not present in the manuscript.
- Legend Fig.2- there was some grammatical errors- independent, experiments... Please review.
- Fig 4 Legend. Review considering that there was errors: e.g., Fig. S9 was not present (probably refer to Fig S8), Table 3 was not present in the manuscript.
- Fig 5 Legend. Please review considering that there is some errors.
- Fig.2a Legend. Authors must include reference to the control used PMM25(ZEO).

Reviewer #2:

Remarks to the Author:

Bhattacharya et al from the group of Marc Ouellette build on their previous non-assumptive approach using Cos-Seq and employ Mut-Seq to identify candidate loci that confer drug resistance to various chemotherapeutic agents used to reduce the burden of disease caused by Leishmania parasites. Drug resistance is an emerging issue and novel strategies to identify resistance loci are needed, and the authors provide a compelling dataset illustrating their successful use of forward chemical mutagenesis screens coupled with NGS (often referred to as Mut-Seq) to identify gain-of-function SNV mutations in genes that confer drug resistance. Their approach is compelling and proves that strategies used successfully to identify resistance loci in haploid organisms can be readily applied to Leishmania, and likely other organisms of various ploidy (diploid-to-aneuploid).

There are many interesting aspects to their study. First, they optimized the dosage and type of chemical mutagen used to select for PMM and MIL resistance. All worked, but with slightly different efficiencies. It would be nice if the authors were more explicit on the dosage that is best because the two dosages used had about a log difference in SNVs (SNPs and INDELS) – which is preferable? For PMM, it was the lower dose that provided the majority of targets based on their selection criteria, whereas for MIL, it was the higher dose? While I agree with their overall selection criteria, it rather seems equally logical to prioritise candidates that are pulled out in both high and low drug pressure (independent screens) in the MIL treatment, for example LinJ 07.0410 protein kinase, putative. Is this part of the same CDPK family as CDPK1? Likewise, candidate genes that are pulled out in both MIL and PMM screens, such as LinJ 20.120 hypothetical protein, as this may serve as a more general target. Out of curiosity, what was the rationale for not testing the above two candidates in their reversion phenotype screen, as this seems an obvious oversight.

Second, their follow-up on multiple drug targets for their reversion phenotype, with the exception of the two mentioned above, was quite thorough and convincing, including using gene editing to revert one mutant and establish that the SNV is responsible for the phenotype, and not due to some off-target mutation. The main thrust of the paper is to attempt to dissect the precise molecular basis for how the CDPK1 mutant allele confers drug resistance. They present good evidence that the CDPK1 mutant alters protein translation and protein folding. I did not find any discussion on the role CDPK1 plays in other protozoan parasites, specifically in calcium-regulated secretion and artemisinin resistance in *Toxoplasma gondii*. My understanding is that in other published PMM resistant mutants, antagonists of Ca²⁺ channels increases PMM susceptibility – could a similar process be operating here? Does incubation with either verapamil or amlodipine, antagonists of Ca²⁺ channels, alter the PMM resistance in the CDPK1 mutant? Or for that matter, does directly inhibiting CDPK1 restore PMM sensitivity?

Further, is the Leishmania CDPK1 reported here directly orthologous to the Toxoplasma CDPK1. Toxoplasma expresses about 20 related CDPKs, a quick BLASTp search suggests otherwise, but one of the top hits is the Leishmania CDPK1, LinJ 33.1810. How big is the CDPK family in Leishmania? Given the interest in developing antagonists for CDPK1 in Toxoplasma and Plasmodium, inclusion of some discussion how the Leishmania results inform or contrast with the work in the apicomplexa seems warranted.

Lastly, several projects have reported using WGS on natural isolates that are resistant to antimonials and other anti-Leishmania drugs. It would be great to highlight whether any of the target genes identified herein have emerged in natural populations. Likely this has not been systematically done. Following up on that, have the authors considered, as proof-of-principle, to generate antimonial mutants to show that their methodology identifies similar mutations in, for example, the aquaglyceroporin gene that was identified by WGS in naturally circulating Leishmania parasites that had developed resistance to antimonials, reported in eLife (2016).

Minor comments:

1. Given the ability of Leishmania to exhibit mosaic aneuploidy during in vitro culture, mentioning that the clones were effectively at passage 1 after selection at the time of drug testing may be good to add to the results. It saves the reader looking to the methods section to find that information
2. Figure 5 legend, it should read "...proteins identified by LC-MS/MS"
3. Figure S3, please change the PMM labels for each clone to PMM1, PMM2, etc., unless the clones are specifically referred to as PPM. In Table S3, they are referred to as PMM. Please clarify.
4. Table S1, it seems more logical for the MIL screen to change the zero's for ENU 4mM and MMS 0.1mM to ND, as these were not done, rather than produced zero colonies. Likewise for the PMM screen, change the HMPA 100mM selection to ND.

Reviewer #3:

Remarks to the Author:

The manuscript 'Coupling Chemical Mutagenesis to Next Generation Sequencing for the identification of drug resistance mutations' by Bhattacharya et al uses a Mut-seq approach to reveal SNPs that are associated with miltefosine and paromomycin resistance in vitro. This approach revealed a considerable number of genes whose heterozygous or homozygous mutations correlated with drug resistance, a possibility that was confirmed for a selection of genes by partially restoring drug susceptibility when expressing a wildtype copy of the gene of interest. The authors then focus on one candidate drug resistance gene, the protein kinase CDPK1, conducting a series of control experiments to link mutation of this kinase and drug resistance and exploring its potential interactions. Overall this study is largely descriptive in nature and the specificity of the results (i.e. link between mutation and drug resistant phenotype) has not been established convincingly. Furthermore, analyses conducted with CDPK1 are poorly controlled and the results were not validated experimentally.

Major comments:

- 1) Even though the authors attempt to proof the specificity of their assay by over-expression of wt copies, this often results in only partial restoration of the original drug susceptibility phenotype. This is an indication that the phenotypes are not just simply due to the mutation but that other, confounding genetic factors are in play, including clonal variations in karyotype (which has been documented in a supplementary figure), gene copy number variations (not investigated) or the many other mutations observed in each individual clone that may have important consequences on enzymatic activities and

thus the cellular phenotype. Each individual mutation linked to drug resistance therefore needs to be validated in a 'clean' context - i. e. drug resistance phenotypes need to be reproduced by gene editing in a wt parasite. This problem is quite well illustrated by the fact that the strongest genetic signal - a deletion on chr 6 observed in all 25 PMM resistant clones - was not validated by over-expression of the corresponding two ORFs encoded at this locus.

2) The authors focus only on coding sequence even though mutagenesis equally affected inter-genic regions. In line with the comment above this may have important consequences on stability and translational control of certain mRNAs given the known role of Leishmania 5' and 3' UTRs in these processes.

3) In their functional over-expression tests, the authors deliberately focused on genes that were already linked previously to drug resistance, which thus renders this manuscript more confirmatory rather than giving new insight. As a result, for many of the known drug resistance genes they tested with this approach, it is not surprising that the overexpression causes increased drug susceptibility (indeed MT expression levels have been linked to MIL susceptibility before). The overexpression approach is thus not a proof of a specific complementation for drug susceptibility in the mutants.

4) The authors establish the genetic link of CDPK1 to the observed drug resistance phenotype using a series of experiments, including null mutant analysis or reproducing the observed mutation in a wt context using gene editing. Surprisingly, the mutation causes also antimony resistance, demonstrating that the resistance mechanism is generic rather than specific. The link of the mutant phenotype to reduced translation efficiency suggests that these parasites may simply undergo a stress response caused by intrinsic defects that result from altered CDPK1 biological function. As previously shown for MIL resistance, this could involve mitochondrial HSP70 (Vacchina et al 2016) or any other intrinsic stress factor that may have undergone amplification. It is surprising that this was not followed up more closely. What is the status of stress protein expression in these parasites (many antibodies do exist to address this question)? Any stress gene amplification observed? How do these parasites cope with heat, pH or nutritional stress? What is their growth profile (reduced growth and biosynthetic activity can cause reduced drug susceptibility)?

5) To investigate the resistance mechanism more closely, the authors conduct pull down experiments and determine co-precipitated proteins by MS analysis. The specificity of these experiments and the statistics of the results are unclear (table s6 has also a formatting problem). Reciprocal pull down experiments need to be conducted to validate the individual interactions and in vitro kinase assays need to be performed to establish the direct kinase-substrate relationship for the proposed, putative substrates.

6) The discussion simply reiterates the results and provides no convincing argument on the broader impact of this paper.

Reviewer #1 (Remarks to the Author):

-Why authors do not have used the Mut-Seq strategy to identify drug resistant markers for Amphotericin B?

We have applied Mut-seq for antimony and methotrexate and those screens worked well and results are being analysed. We also applied Mut-seq to AMB. However, this is proving more challenging. Following considerable optimisation we finally obtained colonies growing on AMB containing plates (best conditions were 80mM EMS and 15X EC50 AMB). However, those cells were marginally resistant with EC50 only 2X higher than control cells, and not worth the further characterization by NGS. Future additional work and optimisation will be needed for studying AMB.

-Could the authors validated the most relevant identified drug resistant markers in clinical/experimental drug resistant Leishmania isolates? Although, no clinical resistant isolates to PMM have been reported, probably by its low routine use in the field, authors could analyse in experimental PMM resistant lines the selected new markers of PMM resistance (specifically the protein kinase CDPK1).

We have analysed the genome of conventionally raised mutants and we found mutations in CDPK1 and the 50S ribosome-binding GTPase in PMM resistant mutants, and mutations in MT1 and the fatty acid elongase in MIL resistant mutants. Two distinct approaches (Mut-seq here and NGS of drug resistant mutants) have thus converged on the same resistance determinants, strengthening the Mut-seq approach. This new important and supporting data have been included in the revised manuscript in the result section (p.8 and 9) and Table 1 was modified for including this information.

-CDPK1 could be linked to MIL and/or AmB resistance?. Considering the involvement of this protein kinase in the PMM and SbIII resistance, it would be interesting that authors demonstrate the possible involvement of this kinase in the resistance to MIL and/or AmB in Leishmania.

Note that in the initial manuscript we had already mentioned (p.11- now p.12) that CDPK1 was not linked to MIL resistance but we had not shown the data, thus that could have been easily missed. This data is now included in Fig. S8c. We also tested AMB but the results were less clear. A CDPK1 single KO was more sensitive (not resistant); and since the phenotype- a 2-fold sensitivity to AMB in the CDPK1 SKO- was not reverted in the add-back we prefer to leave it to MIL.

-Other relevant question concerns about if drug resistant markers identified in *L. infantum* using the Mut-Seq strategy could be generalized for others Leishmania species, or on the contrary for each Leishmania species, different drug resistant markers could be obtained using Mut-Seq. Authors must indicate in the manuscript reference to this point.

This is a good point raised by the reviewer for which we had already given some thoughts. The main resistance genes will probably produce resistance across species; for example mutations in the miltefosine transporter or CDPK1 were shown in both *L. major* and *L. infantum*. But indeed metabolism differs between species and this may lead to species specific resistance. At this point we can only speculate but experiments are planned to

replicate the MIL Mut-Seq screen with *L. major*. This is now mentioned at page 20 of the revised manuscript.

-The EMS, as the most potent mutagen, could have similar effects in different Leishmania species, and consequently could determine different mutations and different drug resistant markers?.

This cannot be excluded but as explained above it awaits further experimental validation.

Minor points:

All the minor points were taken into account and corrected as suggested.

Reviewer #2 (Remarks to the Author):

There are many interesting aspects to their study. First, they optimized the dosage and type of chemical mutagen used to select for PMM and MIL resistance. All worked, but with slightly different efficiencies. It would be nice if the authors were more explicit on the dosage that is best because the two dosages used had about a log difference in SNVs (SNPs and INDELS) – which is preferable? For PMM, it was the lower dose that provided the majority of targets based on their selection criteria, whereas for MIL, it was the higher dose?

We believe that the reviewer refers to Table S1. Only one concentration of mutagens is shown for EMS, ENU, MMS or HMPA. Using more or less of the mutagens in combination with MIL or PMM was not leading to colonies and this was already mentioned in the discussion. Different mutagens induced different types and number of mutations and this was already acknowledged in the original version. However we concentrated on cells that were selected at the highest drug concentration of MIL or PMM. Since we had enough clones for PMM at 10X the EC50 we focussed on those 25 clones. For MIL we only had 3 at 10X so this explains why we picked colonies (13/16) at 5X EC50. Table S1 was re-formatted to avoid further confusion.

No difference were observed between the two MIL concentration groups in terms of number or type of mutations (Fig. S3, MIL1-7 vs MIL8-10). The ideal condition would be the mutagen with the less SNPs but leading to several clones highly resistant to either MIL or PMM. In this regard clones mutagenized with HMPA or MMS have at least 10-fold less SNPs, than clones mutagenized with EMS and yet have resistance levels similar for PMM (Fig. 1). Thus on this aspect one could say that HMPA or MMS are 'better' but they give rise to less mutants and we believe that different mutagens are valuable since we are looking at convergence of mutations. This is now briefly discussed at page 16.

While I agree with their overall selection criteria, it rather seems equally logical to prioritise candidates that are pulled out in both high and low drug pressure (independent screens) in the MIL treatment, for example LinJ 07.0410 protein kinase, putative. Is this part of the same CDPK family as CDPK1? Likewise, candidate genes that are pulled out in both MIL and PMM screens, such as LinJ 20.120 hypothetical protein, as this may serve as a more

general target. Out of curiosity, what was the rationale for not testing the above two candidates in their reversion phenotype screen, as this seems an obvious oversight.

We did not use MIL concentrations as a criteria for gene prioritisation. Indeed we had no indication that it influenced the number or type of mutations and the mutation profiles of resistant mutants within each screen were largely different. The mutated genes were thus analysed on the basis of the recurrence in both MIL concentration used for selection.

Also, we found that mutation diversity helps in predicting genes likely to have a role in resistance. In the case of LinJ.20.1200, the same mutation was seen in all mutants, making it of less interest. Of note this gene was solely found in MIL mutants, it is not shared with PMM mutants. When looking at genes shared by MIL and PMM mutants (i.e. genes shaded in gray in Tables S2 and S3) we noticed that some of the mutations had low quality scores. These have now been analysed by conventional sequencing and mutations in genes LinJ.07.0410 and LinJ.03.0410 (the most interesting based on function) came out to be false-positive bioinformatics artefacts. These have been removed from Tables S2-S5. This explains (identical mutation in all mutants or low quality score) why we did not work with the two genes listed by reviewer 2.

They present good evidence that the CDPK1 mutant alters protein translation and protein folding. I did not find any discussion on the role CDPK1 plays in other protozoan parasites, specifically in calcium-regulated secretion and artemisinin resistance in *Toxoplasma gondii*. My understanding is that in other published PMM resistant mutants, antagonists of Ca²⁺ channels increases PMM susceptibility – could a similar process be operating here? Does incubation with either verapamil or amlodipine, antagonists of Ca²⁺ channels, alter the PMM resistance in the CDPK1 mutant? Or for that matter, does directly inhibiting CDPK1 restore PMM sensitivity?

CDPKs are being studied in Apicomplexa and since next point raised by reviewer 2 relates to this we have discussed similarities and differences between the *Leishmania* and Apicomplexa proteins (see pages 9 and 18).

Related to calcium and paromomycin and CDPK1 we were aware of the data of the Salotra lab (IJPDDR 2017) showing that amlodipine reduces PMM drug resistance in PMM resistant isolates. We confirmed that amlodipine decreased by 10-fold the PMM susceptibility in wild-type cells and it reduces PMM resistance in the mutant PMM25 where CDPK1 is mutated also by 10-fold. This does not seem related to CDPK1. It has not been possible to study this in our knockout cell since unexpectedly the HYGRO marker present in our recombinant parasites (for the replacement of one *CDPK1* allele) produced high resistance to amlodipine.

Further, is the *Leishmania* CDPK1 reported here directly orthologous to the *Toxoplasma* CDPK1. *Toxoplasma* expresses about 20 related CDPKs, a quick BLASTp search suggests otherwise, but one of the top hits is the *Leishmania* CDPK1, LinJ 33.1810. How big is the CDPK family in *Leishmania*? Given the interest in developing antagonists for CDPK1 in *Toxoplasma* and *Plasmodium*, inclusion of some discussion how the *Leishmania* results inform or contrast with the work in the apicomplexa seems warranted.

In this revised version we have expanded on the relationship of CDPKs in the discussion at page 18. A phylogenetic tree (Fig. S4b) was already provided and this has been expanded (new Fig. S4b). There are only 2 *Leishmania* CDPK and they cluster together, with their *Trypanosoma* (a related parasite) orthologues (Fig. S4b).

Lastly, several projects have reported using WGS on natural isolates that are resistant to antimonials and other anti-*Leishmania* drugs. It would be great to highlight whether any of the target genes identified herein have emerged in natural populations. Likely this has not been systematically done. Following up on that, have the authors considered, as proof-of-principle, to generate antimonial mutants to show that their methodology identifies similar mutations in, for example, the aquaglyceroporin gene that was identified by WGS in naturally circulating *Leishmania* parasites that had developed resistance to antimonials, reported in eLife (2016).

As indicated in our answer to reviewer 1, we have shown that some of the mutations highlighted in this study were also shown in mutant selected for drug resistance in a stepwise fashion. This validates the Mut-seq approach. The analysis of a screen with antimony is ongoing; we have already provided a snapshot of it showing that CDPK1 is mutated (Fig. S8d), preliminary data are showing that in 3 mutants derived from Mut-seq we observed a mutation in the aquaglyceroporin gene AQP1. While preliminary and clearly not extensively studied it is possible that some of the phenotypic mutations shown here may find their equivalent in genes emerging from natural populations.

Minor comments:

All the minor points (except point 4) were taken into account and corrected. For point 4, the experiments were carried out and the '0' indicated that indeed under the conditions tested there were 0 colonies.

Reviewer #3 (Remarks to the Author):

The manuscript 'Coupling Chemical Mutagenesis to Next Generation Sequencing for the identification of drug resistance mutations' by Bhattacharya et al uses a Mut-seq approach to reveal SNPs that are associated with miltefosine and paromomycin resistance in vitro. This approach revealed a considerable number of genes whose heterozygous or homozygous mutations correlated with drug resistance, a possibility that was confirmed for a selection of genes by partially restoring drug susceptibility when expressing a wildtype copy of the gene of interest. The authors then focus on one candidate drug resistance gene, the protein kinase CDPK1, conducting a series of control experiments to link mutation of this kinase and drug resistance and exploring its potential interactions. Overall this study is largely descriptive in nature and the specificity of the results (i.e. link between mutation and drug resistant phenotype) has not been established convincingly. Furthermore, analyses conducted with CDPK1 are poorly controlled and the results were not validated experimentally.

Major comments:

1) Even though the authors attempt to proof the specificity of their assay by over-expression of wt copies, this often results in only partial restoration of the original drug susceptibility phenotype. This is an indication that the phenotypes are not just simply due to the mutation but that other, confounding genetic factors are in play, including clonal variations in karyotype (which has been documented in a supplementary figure), gene copy number variations (not investigated) or the many other mutations observed in each individual clone that may have important consequences on enzymatic activities and thus the cellular phenotype. Each individual mutation linked to drug resistance therefore needs to be validated in a 'clean' context - i. e. drug resistance phenotypes need to be reproduced by gene editing in a wt parasite. This problem is quite well illustrated by the fact that the strongest genetic signal - a deletion on chr 6 observed in all 25 PMM resistant clones - was not validated by over-expression of the corresponding two ORFs encoded at this locus.

We have worked extensively in the past with antimonial resistance in *Leishmania* and shown conclusively that resistance is multifactorial with many genes involved. In this Mut-seq approach we have several SNPs (from 10 to 670 in each mutant) and it is expected that in a single mutant more than one gene can potentially contribute to resistance. In this first use of Mut-seq in *Leishmania* we were interested in finding as many genes associated with a drug resistance phenotype. We posited that transfection of the WT gene in a mutant was the best strategy to study the role of several genes in the drug phenotype. So obtaining 'only partial restoration of the original drug susceptibility phenotype' was exactly what we had expected. This is now explained on p. 17. We of course agree with reviewer #3 that the phenotypes are due to a collection of mutations. We believe that these mutations are most likely other SNPs (a reasonable hypothesis because we used chemical mutagenesis).

We have looked for CNV -so it was studied- and only detected the deletion of 4 genes in chromosome 6 that will be discussed below. No other recurrent CNVs were observed and this was in the original submission. Changes in ploidy were reported (Fig S1) but this is very frequent in *Leishmania* and it remains to be proven that these changes are leading to changes in phenotypes (so far ploidy has been correlated to RNA levels in *Leishmania*, but not more as far as we know (Iantorno et al., 2017 mBio)). Many of the changes in ploidy described here were with the same chromosomes with the two drugs suggesting that the mere plating of the parasite may also impact ploidy. The reconstruction of resistance by adding one by one the mutations by editing different genes is certainly a laudable goal but this has to be for the long term. Nonetheless just to prove that our approach was valid we edited two additional genes. The first was the fatty acid elongase LinJ.14.0790, never yet associated with resistance to miltefosine. We use the same strategy as for CDPK1 with co-transfection of a ZEO construct targeting PTR1. We succeeded in inserting the mutations in one allele of this elongase and this heterozygous mutant indeed exhibit miltefosine resistance (see text p. 12 and Fig. S7d). The second was the MT gene and inserting the mutation in two alleles of otherwise wild-type parasites increase resistance to miltefosine by ~5 fold (see text p. 12 and Fig. S7d).

We did not believe that the deletion in the 25 PMM mutants in chromosome 6 had an important role in resistance. Our experience has shown that even the most important

resistance mechanisms are not found in all the mutants. Diversity is the rule. Thus when the SAME mutation is present in all the mutants; it is suspected as an artefact but in this case it was studied (due to our long standing interest in ABC proteins). We have shown in the past (Ubeda et al., PloS Biol, 2014) that *Leishmania* continuously rearrange its genome by homologous recombination at the level of repeats. Such an event was probably selected early during the experiment (e.g. during mutagenesis) and explains that all the clones studied had that deletion.

2) The authors focus only on coding sequence even though mutagenesis equally affected inter-genic regions. In line with the comment above this may have important consequences on stability and translational control of certain mRNAs given the known role of *Leishmania* 5' and 3' UTRs in these processes.

This is a valid point. Changes in expression in *Leishmania* related to drug resistance are usually due to changes in CNV, but theoretically we cannot exclude that SNVs in intergenic regions are changing the expression of genes/mRNAs. We had already indicated in the initial submission that intergenic SNVs were present (Fig. S3). We searched for homozygous SNVs in intergenic regions recurring in more than one mutant. Nine were studied, for five out of 9 we had evidence that the SNVs were associated with low differential RNA expression as measured by qRT-PCR (less than 2 fold differential expression). We transfected the gene studied in mutants (for gene downregulated) or in WT (for gene upregulated) and for one (ABCC8) we could find a role in resistance to miltefosine (low but significant). This new information is found on p.8 and p.17 and in a new supplementary Table S6 (former Tables S6-9 are now Tables S7-10).

3) In their functional over-expression tests, the authors deliberately focused on genes that were already linked previously to drug resistance, which thus renders this manuscript more confirmatory rather than giving new insight. As a result, for many of the known drug resistance genes they tested with this approach, it is not surprising that the overexpression causes increased drug susceptibility (indeed MT expression levels have been linked to MIL susceptibility before). The overexpression approach is thus not a proof of a specific complementation for drug susceptibility in the mutants.

We respectfully disagree with this statement. With the exception of MT, which served as a wonderful control, all the other genes were for the first time associated with MIL or PMM resistance. We investigated in great detail one of those new genes (*CDPK1*).

4) The authors establish the genetic link of *CDPK1* to the observed drug resistance phenotype using a series of experiments, including null mutant analysis or reproducing the observed mutation in a wt context using gene editing. Surprisingly, the mutation causes also antimony resistance, demonstrating that the resistance mechanism is generic rather than specific. The link of the mutant phenotype to reduced translation efficiency suggests that these parasites may simply undergo a stress response caused by intrinsic defects that result from altered *CDPK1* biological function. As previously shown for MIL resistance, this could

involve mitochondrial HSP70 (Vacchina et al 2016) or any other intrinsic stress factor that may have undergone amplification. It is surprising that this was not followed up more closely. What is the status of stress protein expression in these parasites (many antibodies do exist to address this question)? Any stress gene amplification observed? How do these parasites cope with heat, pH or nutritional stress? What is their growth profile (reduced growth and biosynthetic activity can cause reduced drug susceptibility)?

As indicated in our answer to reviewer 1, the observed phenotype is not generic (see modified Fig. S8, showing that CDPK1 has no role with miltefosine). We were surprised also with the SbIII phenotype but we believe that this further increases the interest in CDPK1. Maybe this kinase produces resistance to PMM and SbIII by different mechanisms. We had not observed any growth defect in our CDPK1 single knockout but we now formalised this observation by providing growth curves (Fig. S6a) and have carried out a series of new Western blots showing that a number of stress proteins are equally expressed in the WT and CDPK1 mutant upon heat stress (Fig. S6b). These results are now reported on p.11.

5) To investigate the resistance mechanism more closely, the authors conduct pull down experiments and determine co-precipitated proteins by MS analysis. The specificity of these experiments and the statistics of the results are unclear (table s6 has also a formatting problem). Reciprocal pull down experiments need to be conducted to validate the individual interactions and in vitro kinase assays need to be performed to establish the direct kinase-substrate relationship for the proposed, putative substrates.

This was a valid point. The formatting of Table S6 (now S7) has been fixed. We have produce HA tags for 8 putative interactors. These were co-transfected in cells expressing a 2x-TY1-CDPK version. We have HA-immunoprecipitated those 8 proteins and 4 (LinJ06.0590, LinJ11.1110, LinJ34.1620; LinJ34.0210) consistently pull down CDPK1 as determined with a TY1 antibody (Fig. S10). Thus reciprocal pull downs has confirmed interactions of several proteins found in Table S8/S9 with CDPK1. We used LinJ.06.0590-HA encoding ribosomal protein L23a to look at whether it could be phosphorylated by CDPK1. We chose L23a because we had claimed a role of CDPK1 in translation and also because of evidence in the literature of the association between L23a and cross-resistance to PMM (Das, 2013). We co-transfected LinJ.06.0590-HA with CDPK-HA, performed an HA immunoprecipitation and carried out a kinase assay. The data supported that L23a is indeed phosphorylated by CDPK1 (Fig. 5c and Fig. S10).

6) The discussion simply reiterates the results and provides no convincing argument on the broader impact of this paper.

We have remove some reiteration of results and while answering the queries of the three reviewers, we have improved the discussion.

Reviewers' Comments:

Reviewer #1:

Remarks to the Author:

Considering that Authors have fully answered all questions and suggestions, my recommendation is to ACCEPT the manuscript.

This is a relevant manuscript with interesting results obtained after the use of a well-designed, statistically supported and elegant methodology that will contribute significantly to the search of drug resistance markers in trypanosomatids.

Reviewer #2:

Remarks to the Author:

The authors have responded effectively to all comments made during the peer-review, and the manuscript is now acceptable for publication. Michael Grigg

Reviewer #3:

Remarks to the Author:

The authors have conducted an impressive amount of work and added substantial new data, which improved this manuscript. Nevertheless, there are important issues that remain to be addressed.

Major issues:

1) With the corrections it has become clear that the mutagens were applied on 'freshly picked L. infantum clones'. The resistance of the subsequently selected lines is compared to parental wild-type cells. What are these cells exactly? The initial clones used for the mutagenesis (before treatment) or the population used to derive the clones? Given the clonal variability observed in Leishmania this is an important information and needs to be specified in methods. The IC50 of the initial (untreated) clones needs to be determined if the original parent population was used as control, and all experiments in this paper should be controlled for by this matching clonal control.

2) Fig. S1 b shows the karyotype of the resistant clones. Several things are wrong here and need correction: (i) it seems that both plots are inverted as the aneuploidies mentioned in results do not fit the figure (the inversion seems clear looking for tetrasomic chr 31), (ii) the legend does not explain the color code and labels, (iii) assuming that the numbers at the left of the panels refer to the chromosomes, what does the value '0' stand for? Also, while the WT profile is shown for PMM, no WT profile is shown for MIL. This needs to be added.

3) Fig. 2A: Again, what is the WT control: population or clone? This is essential here as the control will have an important impact on the interpretation of the results. Unlike noted in the text, it seems unlikely that this mutation has arisen early during the mutagenesis procedure given that the deletion is observed in all lines generated with the different mutagens (and thus independent experiments). Do the authors imply convergence? It seems more likely that this mutation may have been present in the initial clone used to derive PMM resistant lines (going back to the initial question asked: is the shown control the original clone before mutagenesis or an unrelated population?). Significantly, reintroduction of the two deleted genes does have a statistically significant effect and resensitizes the cells to PMM. It seems therefore likely that this analysis was done with a spontaneous mutant that already was more resistant to PMM due to a spontaneous deletion. How does this fact that a mutant rather than a WT was used affect the entire experiment and subsequent results?

4) I still argue that the re-expression of the WT gene alone is not a convincing control. In the given

context of massive and uncontrolled mutations in the resistant lines, it seems essential to me to at least establish for the various genes that overexpression of the genes that carry the mutation will not lead to resensitization.

5) Fig. 3 a – c: These technical controls should go into supplementary data and some of the key results shown there should go to the main figure filed, such as Fig. S8.

6) The authors went to great length to further investigate and control the activity and interaction of CDPK1 in response to my initial comments. While these efforts are appreciated, there are a few issues remaining. Why is the specificity of the various experiments not categorically controlled for by the kinase dead mutant? This mutant has been generated and applied for experiments shown in Figs. 2, 3, and S8. Why was this essential control not used for the experiments shown in Figs 5C and S10? Unlike stated in the text, the AMARA peptide used for competition is not specific for CDPK1 and could thus interfere with a co-purified kinase. Likewise, cold ATP alone is not a convincing control for specificity.

7) What are all these non-specific bands in the reciprocal pull down given that a specific antibody for the HA tag was used? What were the objective criteria used to label the correct bands by the arrow heads?

8) I disagree with the statement that the discussion has improved. The new additions are sloppy ('a condition with less SNVs', '...mutagens ...allow a sufficient number of mutants...', '...required a hierarchy of mutations...') and discuss very little about potential mechanisms that may underlie the role of CDPK1 in the observed PMM drug resistance phenotype and cross resistance to antimony, and how their interaction map or substrate phosphorylation (L23a for example) fit with this.

Minor comments:

1) Page 6 line 7: The statement that they focus 'mostly on SNVs in coding regions, as Leishmania do not rely on transcriptional control' is misleading as Leishmania relies on post-transcriptional and translational control, both of which are regulated by the UTRs as shown by the excellent work of this very team. Also it conflicts with the new analyses now included in this manuscript.

2) Table S3: In my version the mentioned gray shading was not visible. What does the asterisk in the last colon first row mean? This should be explained in footnote C.

Dear Editor,

Please find enclosed the revised version of our manuscript. We thank reviewers 1 and 2 for accepting the paper and reviewer 3 for acknowledging that the revised version was improved. We were of course disappointed by the outcome but encouraged that it could further be considered. We have looked carefully at the critical comments of the third reviewer; have carried out further experiments when deemed necessary. A new co-author, Dr. Angana Mukherjee was added as an extra wet-lab experimentalist was necessary since the first author Dr. Arijit Bhattacharya is now back to India. You will find below our detailed point-by-point rebuttal. We hope that this revised version will now be found acceptable for publication in Nature Communications.

Sincerely yours
Marc Ouellette

Major issues:

1) With the corrections it has become clear that the mutagens were applied on 'freshly picked *L. infantum* clones'. The resistance of the subsequently selected lines is compared to parental wild-type cells. What are these cells exactly? The initial clones used for the mutagenesis (before treatment) or the population used to derive the clones? Given the clonal variability observed in *Leishmania* this is an important information and needs to be specified in methods. The IC₅₀ of the initial (untreated) clones needs to be determined if the original parent population was used as control, and all experiments in this paper should be controlled for by this matching clonal control.

The EC₅₀s were calculated from the initial clone used for the mutagenesis. This clone was maintained in parallel with the mutants. This is now clarified in the "methods" section at pages 22 and 23.

2) Fig. S1 b shows the karyotype of the resistant clones. Several things are wrong here and need correction: (i) it seems that both plots are inverted as the aneuploidies mentioned in results do not fit the figure (the inversion seems clear looking for tetrasomic chr 31), (ii) the legend does not explain the color code and labels, (iii) assuming that the numbers at the left of the panels refer to the chromosomes, what does the value '0' stand for? Also, while the WT profile is shown for PMM, no WT profile is shown for MIL. This needs to be added.

Reviewer 3 is right; we apologize for this. With the many versions of figures and draft this went unnoticed when we made our submission. It was indeed inverted. The '0' was an offset for chromosome number. The color code was explained by an inset but we now made a much improved and clearer version of the figure.

3) Fig. 2A: Again, what is the WT control: population or clone? This is essential here as the control will have an important impact on the interpretation of the results. Unlike noted in the text, it seems unlikely that this mutation has arisen early during the mutagenesis procedure given that the deletion is observed in all lines generated with the different

mutagens (and thus independent experiments). Do the authors imply convergence? It seems more likely that this mutation may have been present in the initial clone used to derive PMM resistant lines (going back to the initial question asked: is the shown control the original clone before mutagenesis or an unrelated population?). Significantly, reintroduction of the two deleted genes does have a statistically significant effect and resensitizes the cells to PMM. It seems therefore likely that this analysis was done with a spontaneous mutant that already was more resistant to PMM due to a spontaneous deletion. How does this fact that a mutant rather than a WT was used affect the entire experiment and subsequent results?

The clone from which all mutants were derived was sequenced and it does have the locus on chromosome 6 intact. This was already shown in Fig.S2 (WT) but now this is indicated in the legend of Fig. S2.

Possibly this locus rearranges at a greater rate than others and since it gives some minimal advantage (1.4-fold) in the presence of PMM, it is selected for. As indicated in our previous rebuttal we do not believe that this is a locus of more importance than others in producing resistance to PMM. Supporting the notion that this locus rearranges frequently we also noticed the same deletion in MIL5. MIL 5 has minimal cross-resistance to paromomycin (~1.5X). This info is now included in the manuscript on p.6.

4) I still argue that the re-expression of the WT gene alone is not a convincing control. In the given context of massive and uncontrolled mutations in the resistant lines, it seems essential to me to at least establish for the various genes that overexpression of the genes that carry the mutation will not lead to resensitization.

There is obviously a divergence of opinion here. I still argue that re-expression of a WT copy in a mutant is valid to look at the function of genes in resistance. It is a strategy that we used and published extensively and it is even stronger when it provides a phenotype in 'the context of massive and uncontrolled mutations'. Nonetheless, to address reviewer's point we carried out transfection of the mutated versions for 6 genes (three for each drug). None resensitized mutants to either MIL or PMM, and one mutated gene (LinJ36.6220) even produced more resistance to MIL. This is now included in Table 1 and mentioned at pages 8 and 10.

5) Fig. 3 a – c: These technical controls should go into supplementary data and some of the key results shown there should go to the main figure filed, such as Fig. S8.

We believe that providing the blot supporting that CDPK1 is likely to be essential is more than a technical control. Fig. 3a-b thus remain as mainstream figures, but Fig. 3c is now in Fig. S5a.

We are fine including Fig. S8 as a new main figure and it is now Fig.4 (original Figs 4 and 5 become Figs 5 and 6, respectively and original Figs S9 and S10 become Figs S8 and S9, respectively).

6) The authors went to great length to further investigate and control the activity and

interaction of CDPK1 in response to my initial comments. While these efforts are appreciated, there are a few issues remaining. Why is the specificity of the various experiments not categorically controlled for by the kinase dead mutant? This mutant has been generated and applied for experiments shown in Figs. 2, 3, and S8. Why was this essential control not used for the experiments shown in Figs 5C and S10? Unlike stated in the text, the AMARA peptide used for competition is not specific for CDPK1 and could thus interfere with a co-purified kinase. Likewise, cold ATP alone is not a convincing control for specificity.

Demonstrating that the AMARA peptide was phosphorylated by CDPK1 and that the peptide competes with the phosphorylation of L23a when CDPK1 was immunoprecipitated made us believe that this was sufficient to prove the interaction between the kinase and one of its substrate. These were the tools that were available at the time. The use of peptide to control the activity/specificity of the kinase is well established. However, to address the reviewer's point we co-transfected HA-dead kinase and HA-L23a and carried an in vitro kinase assay and showed that indeed an active CDPK1 is needed to phosphorylate L23a. This is now included in Fig. S9e.

7) What are all these non-specific bands in the reciprocal pull down given that a specific antibody for the HA tag was used? What were the objective criteria used to label the correct bands by the arrow heads?

The nonspecific bands can be due to cross reactivity of the anti-HA or secondary antibodies. The observation of non-specific bands is not uncommon when overexpressing proteins. The bands were marked if they fit the expected molecular weight and are absent in control sample.

8) I disagree with the statement that the discussion has improved. The new additions are sloppy ('a condition with less SNVs', '...mutagens ...allow a sufficient number of mutants...', '...required a hierarchy of mutations...') and discuss very little about potential mechanisms that may underlie the role of CDPK1 in the observed PMM drug resistance phenotype and cross resistance to antimony, and how their interaction map or substrate phosphorylation (L23a for example) fit with this.

We have further improved the discussion.

Minor comments:

1) Page 6 line 7: The statement that they focus 'mostly on SNVs in coding regions, as Leishmania do not rely on transcriptional control' is misleading as Leishmania relies on post-transcriptional and translational control, both of which are regulated by the UTRs as shown by the excellent work of this very team. Also it conflicts with the new analyses now included in this manuscript.

We of course agree that Leishmania uses UTRs for regulating gene expression (e.g. during different life stages) but this is less clear for resistance. We rephrased this sentence (now on p. 8) but I was very surprised by the result.

2) Table S3: In my version the mentioned gray shading was not visible. What does the asterisk in the last colon first row mean? This should be explained in footnote C.

The grey shading is now visible. Footnote has been amended.

Reviewers' Comments:

Reviewer #3:

Remarks to the Author:

The authors have responded to all my queries and the manuscript is now suitable for publication.